

# A physical model for mean river discharge calculation:
# from riverside seismic monitoring experiments in a low-
# flow river, China
Xiaoyue Zhou [1], Liang Feng [1,2], Shizhe Zhang [3], Bin Xie [4], Yanmei Wang [1], Wei Xu [1],
Emanuele Intrieri[5]
1. Faculty of Resource and Environmental Engineering, Jiangxi University of Science and Technology,
Ganzhou, China
2. Jiangxi Provincial Key Laboratory of Water Ecological Conservation in Headwater Regions,
Ganzhou, China
3. Institute of Mountain Hazard and Environment, Chinese Academy of Sciences, Chengdu, China
4. Ganzhou Earthquake Monitoring Center Station,Jiangxi Provincial Seismological Bureau,
Ganzhou, China
5. Department of Earth Science, University of Florence, Florence, Italy
*Correspondence to:* Feng Liang(liang.feng@jxust.edu.cn)
**Abstract.**
The dynamics of water flow and sediment transport in river systems play a crucial role in shaping river
morphology, in the planning and use of river infrastructure and the broader watershed management.
However, these characteristics are often challenging to measure comprehensively. On March 17, 2023,
we studied a low-flow river system (≤0.611 m³/s) within the boundaries of Yuancun in the Township of
Meishui. By synchronously monitoring the microseismic signals generated by the river and the river flow
velocity, we explored the relationship between these microseismic signals and the river discharge. During
each experiment, we used 3 to 4 three-component seismometers placed in close proximity to the
riverbank (at the distance of approximately 1 meter), with one device submerged underwater to record
the microseismic signals caused by the flow. The signals exhibited a wide frequency range (2–50 Hz).
An analysis of the recorded microseismic signals and the flow data revealed an approximate linear
relationship between the seismic noise in the 2–10 Hz bandwidth and the river flow. We used a least
squares regression model to invert the river flow from the 2–10 Hz microseismic signals and found that
the maximum relative error between the inverted flow and the measured values was 10.3%. The results
show that even at low flow rates, real-time monitoring of river processes is possible through continuous
time-frequency analysis of microseismic signals; this increases the potential for future applications of
seismic monitoring in real-time observation of hydrological evolution in river systems.





## 1 Introduction

The components of river monitoring usually include encompass river water level, flow velocity, discharge, and sediment flux. Currently, there are two main approaches to river monitoring: one is the hydrological monitoring system based on hydrological stations, and the other is the remote sensing monitoring method, which has gradually developed with the maturity of remote sensing technology. Both methods require the establishment of hydrological stations and the installation of measuring instruments, which makes the monitoring process complex, time-consuming, and resource-intensive (Roth et al.,2016; Cook et al.,2022; Larose et al.,2015). There is also a risk of the instruments being damaged and data being lost during floods. In recent years, an increasing number of researchers have explored the correlation between microseismic signal fluctuations and changes in river discharge and sediment flux, demonstrating the potential of microseismic monitoring technology for hydrological studies (Turowski et al., 2011). Since the early 1990s, when Govi et al. (1993) first deployed short-period seismometers in river channels to record microseismic signals and investigate the relationship between these signals, river discharge, and sediment transport, more researchers have begun to investigate the relationship between microseismic signals and river hydrodynamics (Rickenmann et al., 2012). Schmandt et al. (2013) have found that the sources of the microseismic signals could be related to water flow noise caused by turbulence, sediments impinging on the riverbed, or acoustic waves generated by interactions between water and the atmosphere. Díaz et al. (2014) conducted 36 months of continuous microseismic monitoring of the Aragon River and identified three types of river-induced seismic events, each with distinct characteristics related to floods caused by moderate rainfall, seasonal snowmelt, and severe storms. This demonstrates that microseismic monitoring can not only aid in studying the hydrological characteristics of rivers but also has significant potential in assessing hydrological hazards. Therefore, continuous microseismic monitoring of ambient noise generated by rivers can provide valuable insights into the study of the hydrological characteristics of rivers.

Microseismic monitoring of rivers can provide a wealth of seismic data on the vibrations of river sediments, and the interpretation of microseismic signals to infer hydrological parameters is one of the essential tasks of microseismic monitoring in rivers. Roth et al. (2016) used broadband (5–480 Hz) microseismic data, river discharge data, precipitation data, and bedload data from the Erlenbach River in



the Swiss Pre-Alps to propose a simple, empirically adjusted linear model for estimating bedload
transport rates. Their predictions showed a strong correlation with the transport rates determined by
calibrated seismic detectors in the river. Burtin et al. (2011) analyzed a braided river with a discharge
range of 1–5 m³/s at low water levels by simultaneously measuring river discharge, bedload, and seismic
signals during a limited summer period. Their analysis revealed that seismic signals in the 1–10 Hz band
were most indicative of changes in the river water level, with seismic waves originating from turbulence.
This demonstrates that microseismic monitoring can be used to investigate hydrological characteristics
even at low discharge and can provide earlier indications of flow changes in the downstream main stem
of the river (Anthony et al., 2018; Gaeuman et al.,2014).

Microseismic monitoring has practical applications in flood monitoring and early warning systems and
could also be used in the future to monitor geological disasters through seismic networks. Analysis and
prediction of geological hazards based on microseismic monitoring offers a significant advantage over
current methods based on hydrological monitoring stations and remote sensing. By analyzing the time-
frequency characteristics of microseismic signals, floods can be identified and their evolution monitored.
Additionally, through inversion methods, hydrological data such as river discharge and sediment content
can be derived. Microseismic technology offers a new method for online monitoring of river dynamics
and flood early warning, which has enormous potential for the assessment of hydrological hazards.

This study focuses on the monitoring of tributaries with low discharge. The Jiuqu River in Yuan Village,
Meishui Township, Shangyu County serves as the research object, and through experimental field studies,
a microseismic monitoring system is deployed along the riverbank of the tributary to monitor the ground
vibrations caused by changes in the flow to stimulate the low-frequency microseismic signals, elaborate
and interpret the river microseismic signals by removing the noise of human activities, such as vehicles,
from the ambient noise, analyse the physical characteristics of the river microseismic signals, and
construct a mathematical model using microseismic signals to invert river flow, predicting real-time river
flow (Viparelli et al.,2011), and providing a reference for monitoring and early warning of river flooding
and downstream river flow changes in the region.



## 2 Experiments

### 2.1 Experiment sites

The river studied in this study, the Jiuqu River, is a tributary of Meishui River, located in the territory of Meishui Township in Shangyou County, China (Figure 1). Meishui Township is situated in a hilly and mountainous area with an altitude of 200-300 m, and its relative height is 50-100 m. The exposed strata are the Devonian and Carboniferous of the Late Paleozoic, and the lithology mainly consists of quartz conglomerate, quartz sandstone, siltstone and dolomitic greywacke. It slopes from north to south, with a gentle terrain, and belongs to the subtropical monsoon climate zone, with an average annual precipitation of 1,235.6 mm. In this study, four monitoring experiments were conducted at four sections of the Jiuqu River with different discharge. Current meters and seismic stations were installed on the riverbank to measure the flow velocity and seismic ambient noise in each segment. For this experiment, we selected a curved section of the Jiuqu River, approximately 1.4 kilometers long, with a river width ranging from 3 to 9 meters and a depth of 0.1 to 0.4 meters. The overall morphology of the river channel resembles that of a drainage canal, with the riverbed consisting of gravel, fine sand, and pebbles. The gravel particle size varies; the upstream section features smaller gravel particles, while the downstream section exhibits larger gravel particles (Aderhold et al., 2015). The maximum gravel size is $50 \times 36 \times 20$ cm, with an average size of $13 \times 10 \times 5$ cm. Some segments of the riverbed contain silt. During the experiment, water samples were taken to measure the sediment concentration, which was found to be approximately 0.5% in the studied river section. Throughout the experimental period, the river's discharge was less than 5 m³/s, which classifyies it as a low-flow river (Figure 2).

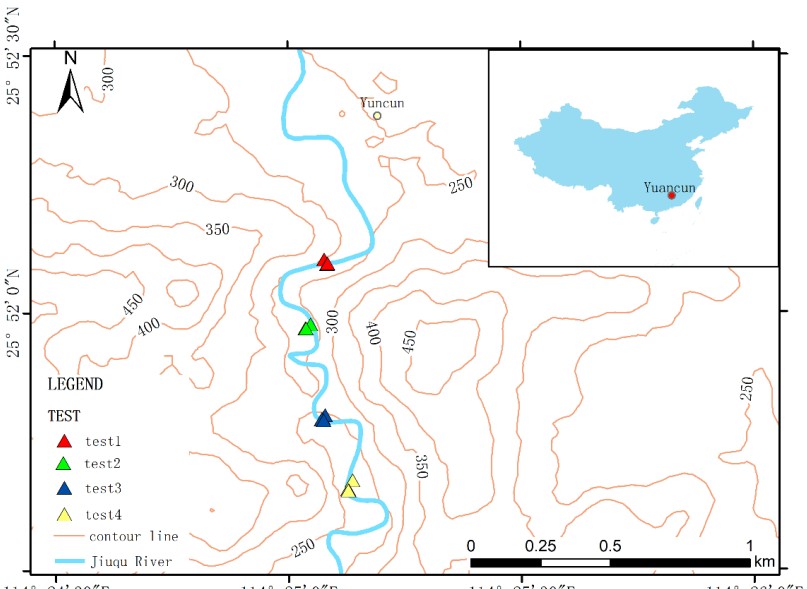

**Figure 1. The geophones at the four experimental sites. The red triangles represent the three base stations in test 1, the green triangles represent the four stations in test 2, the blue triangles represent the four stations in test 3, and the yellow triangles represent the four stations in test 4.**

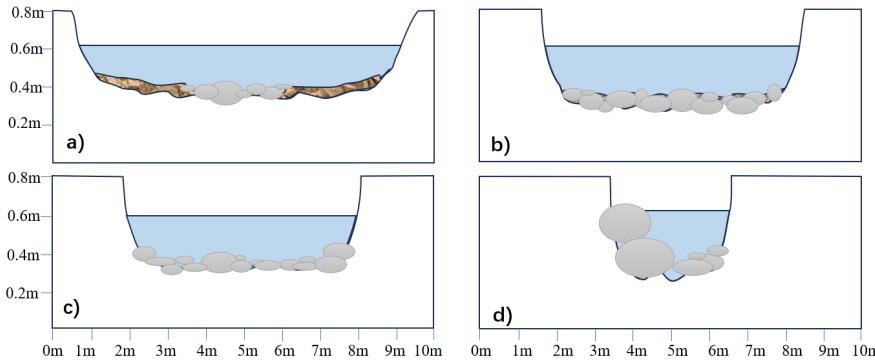

**Figure 2. Cross-sectional views of the river; a), b), c), and d) are cross-sectional views of the river at the first, second, third, and fourth experimental sites, respectively. The grey ovals represent river bottom pebbles, the blue blocks represent river water, and the brown blocks represent river bottom sediment.**

**2.2 Seismic monitoring**

Seismic ambient noise was collected from both the river sections and the nearby road areas. Seismic instruments offer a variety of sensors with different characteristics, such as accelerometer, velocimeter



with different Eigenfrequency and useable frequency bands, making them suitable for specific
applications such as seismic imaging, monitoring, or civil engineering. The instruments can be broadly
categorized into two types: short-period seismometers and geophones, which are sensitive to high-
frequency bands ranging from 1 to 10 Hz up to 500 Hz. These instruments are well-suited for monitoring
most geomorphological seismic sources. Surface processes also generate low-frequency seismic waves,
requiring the use of broadband seismometers, which are sensitive to higher frequency signals as well.
Both broadband and most short-period seismometers typically record seismic signals in three-
dimensional configurations across orthogonal axes aligned vertically, north-south, and east-west,
facilitating the comparison of directional information from multiple stations. The use of three-component
instruments allows for greater data processing diversity compared to single-component instruments. For
instance, the polarity of seismic waves can be used to determine the source type and provide information
about the direction of wavefront incidence in localization problems (Burtin et al., 2014).

The seismic stations used in this experiment were four three-axis velocimeter stations (S45 triaxial
velocimeter and SL06 recorder, SARA electronic instruments s.r.l., Italy), which was utilized to monitor
seismic signals generated by the river, with a sensitivity factor of 78 V/m/s. The sensor components of
the device were east-west (E), north-south (N), and vertical (Z). The natural frequency of these
instruments was 4.5 Hz, and we set the sampling frequency for all instruments to 200 Hz. A total of four
seismic stations were employed, one of which (Station 3) was an integrated velocimeters and data
collector (Velbox, SARA electronic instruments s.r.l., Italy). The other three stations were separate, each
equipped with a SARA 24-bit A/D converter (SL06), connected to a 24-bit digitizer via the converter.
Each monitoring device was leveled using a spirit level and placed on a triangular support base to isolate
it from the ground. Real-time positioning of each monitoring device was conducted using GPS, and the
power supply for the monitoring devices was provided by outdoor 12 V-60 A batteries shared between
different stations.



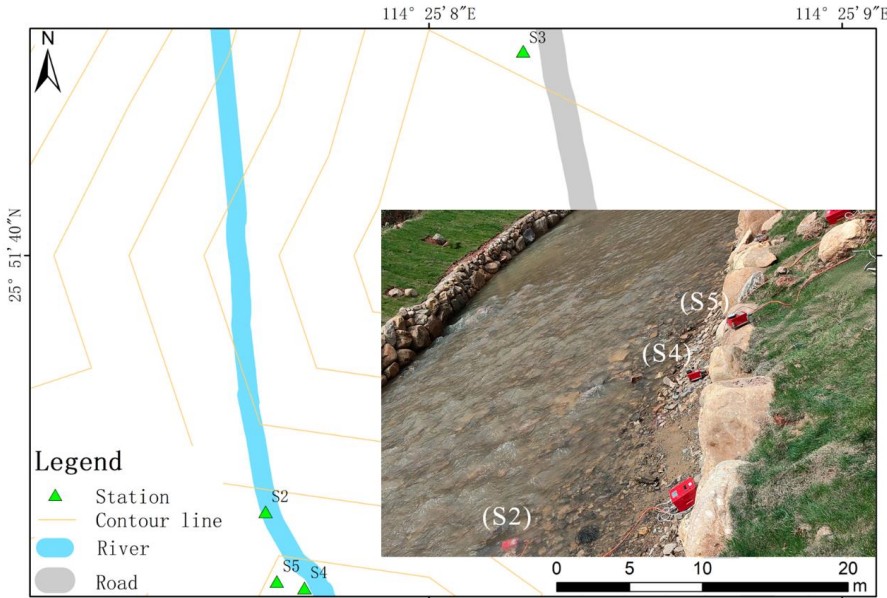


**Figure 3. Detailed location distribution map of the four stations in test2, where S2 was placed in the middle of the river channel to be flooded, S4 was placed 1 m away from the river channel, S5 was placed 1.5 m away from the river channel, and S3 was placed 1.5 m from the road.**

During the research period on March 17, 2024, we conducted four experiments on the Jiuqu River in Meishui Township, measuring seismic data from four river sections. Each experiment lasted for 20 minutes, during which 3 to 4 seismometers were installed approximately 1.5 meters from the riverbank to monitor the seismic signals generated by the water flow. Since the river sections are located adjacent to a road, vehicle and human activities occurred during the experiments. Therefore, in all four experiments, the S3 (Station 3) was placed about 1 meter from the riverbank, near the road. This configuration aimed to record microseismic signals generated by river activities while minimizing interference from human activities.

For the second experiment, the detailed positions of the four stations and the microseismic signals recorded by each station are illustrated in Figure 3. Given that the studied river sections are classified as low-flow segments, the instruments were placed very close to the river channel. Specifically, S2 was submerged in the middle of the river, S4 was located 1 meter from the riverbank, S5 was positioned 1.5 meters away, and S3 was situated 1.5 meters from the road (Figure 3).





By processing the signals recorded by the four stations and plotting the spectrograms, we found that the
dominant frequency range of the microseismic signals generated by the river was between 2 and 10 Hz.
In contrast, the noise frequencies generated by human activities and vehicle traffic concentrated
between 7 and 25 Hz (Figure 4). Moreover, the differences in the arrangement of the stations relative to
the river channel indicated that the stations were unable to monitor detailed microseismic signals at
greater distances from the riverbank. This limitation is primarily due to the nature of the studied river
as a low-flow system, which does not produce sufficiently strong signals.

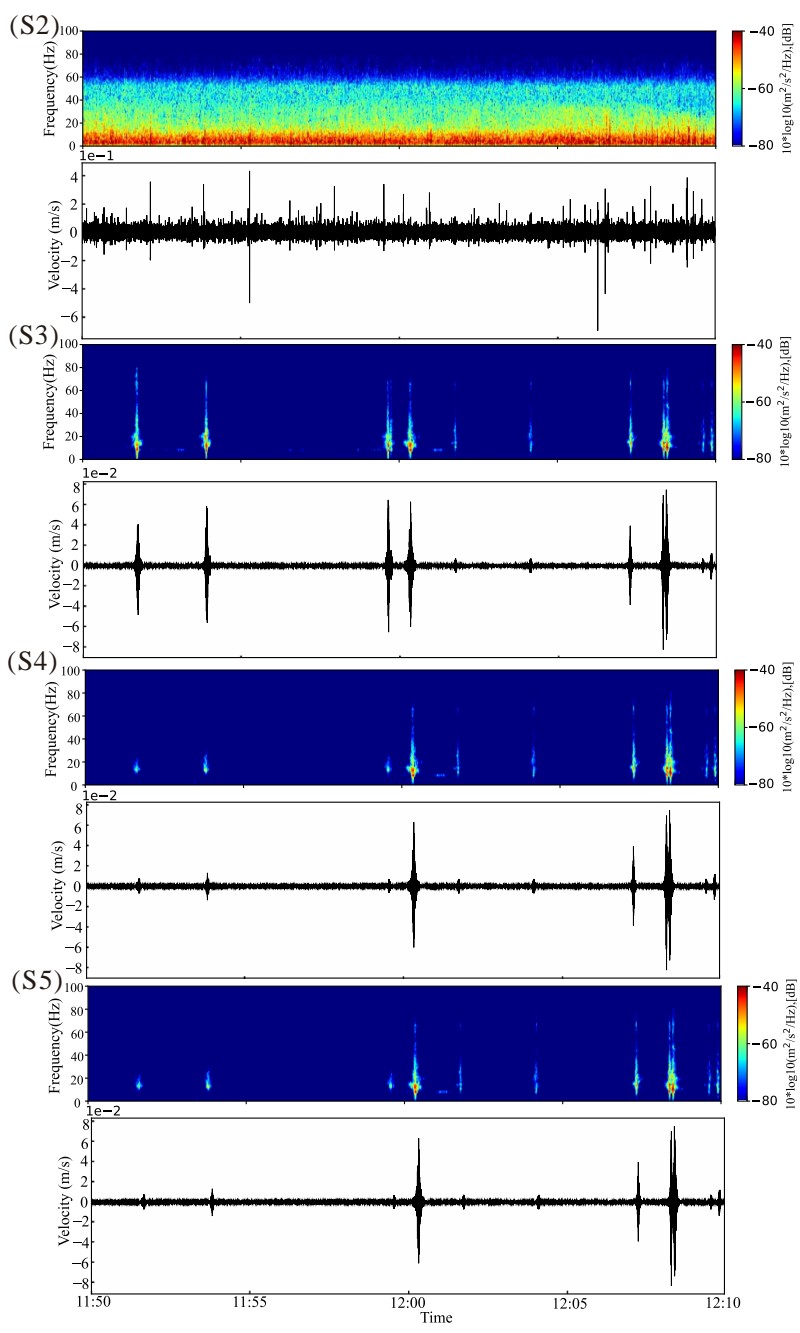

**Figure 4. Seismic waveform and spectra recorded by seismometers, S2, S3, S4, and S5, at the test 2.**

170

172



**2.3 River flow velocity measurement and discharge calculation**

In the study, both the flow velocity and water depth of the river sections were measured. The flow

velocity was measured using a portable flow velocity meter (model LS1206B, Nanjing Jun Can

Instrument Equipment Company, China). To enhance the accuracy of the measured flow velocities,

vertical and horizontal sampling interval were set at 0.1 m and 1 m in a river section, respectively. In

the third experiment, a continuous flow velocity meter was employed to monitor flow velocity over a

period of twenty minutes.

Calculating river flow is typically achieved by determining the average flow velocity of the water

passing through the measured cross-sectional area. Additionally, flow can be directly measured using

appropriate devices or estimated using indirect methods such as empirical equations and mathematical

models. This study utilized a common flow calculation method, the velocity-area method (Herschy,

1993). The principle of this method involves dividing the river's cross-sectional width into several

slices based on the cross-section, then calculating the flow for each slice using its average slice velocity

and slice area, and finally summing these to obtain the total river flow. A schematic diagram of the

calculation is shown in Figure 5.

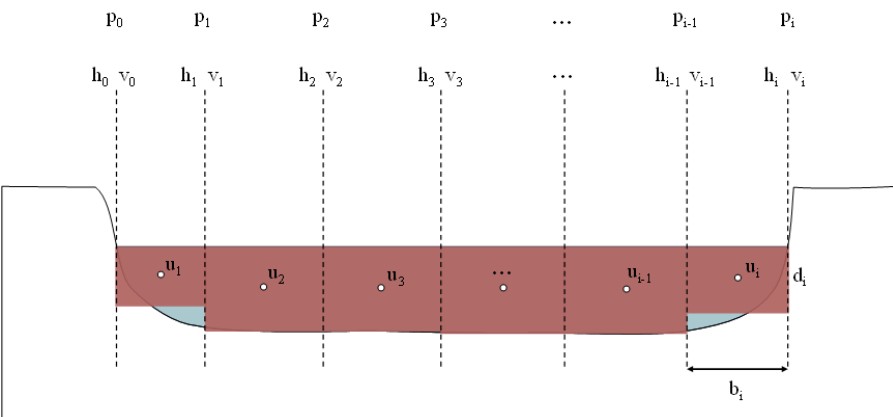

**Figure 5. Schematic diagram of the velocity-area method for estimating river flows**

Calculate the width of the slice $b_i$:



$b_i = p_i p_0 - p_{i-1} p_0$                                                                 (1)
Where $p_i p_0$ denotes the distance from the $i$ th vertical measurement point to the start of the riverbank.
Calculate the depth of the slice $d_i$
$d_i = \frac{h_{i-1} + h_i}{2}$                                                              (2)
Where $h_{i-1}$ and $h_i$ are the water level measured at the $i$-1 vertical measurement point and the $i$ vertical
measurement point, respectively.
Multiplying the result of equation (1) with the result of equation (2) gives the slice area $A_i$:
$A_i = b_i d_i$                                                                              (3)
The most common methods for determining the mean slice velocity (mean vertical velocity) are the
vertical velocity profile method, the two-point method and the six-tenths depth method. Since only the
surface velocity of the river was measured in this experiment and that the study river is a low flow stream,
the linear relationship proposed by (Genç O et al., 2015) twas used for the calculation of the mean slice
velocity. The mean slice velocity $u_i$:
$u_i = 0.552 \bar{u}_{wsi}$                                                                  (4)
Where $\bar{u}_{wsi}$ denotes the slice-averaged water surface velocity and its value is:
$\bar{u}_{wsi} = \frac{v_{i-1} + v_i}{2}$                                                     (5)
Where $v_{i-1}$ and $v_i$ are the surface flow velocity measured at the $i$-1 vertical measurement point and
the $i$ vertical measurement point, respectively.
The slice flow rate for each slice $q_i$ is obtained by combining equation (3) and equation (4):
$q_i = u_i A_i$                                                                              (6)
Final river flow $Q$ obtained:$Q = \sum_{i=1}^{n} q_i$

215    (7)

Where $n$ denotes the number of slices.
The profile data measured in the four experiments were processed by the above calculation process to
obtain the river flow rates of 0.444 m³/s, 0.611 m³/s, 0.512 m³/s, and 0.598 m³/s for tests 1, 2, 3, and 4,
respectively.





### 3 Seismic ambient noise

Geophones can detect elastic waves generated by processes occurring on or near the Earth's surface, which are emitted by the transmission of energy from objects impacting the ground, such as boulders falling from slopes, pebbles bouncing on the riverbeds, or raindrops falling to the ground. The sources of elastic waves generated by river processes can be quite complex and depend on the flow configuration of the river. Possible sources include particle collisions during sediment transport, turbulence, bubble explosions, friction between water flow and riverbed or riverbanks, and the propagation of gravity waves or breaking waves on the river surface. The frequency range and physical characteristics of microseismic signals produced by different river processes are distinct. By analyzing images such as time-frequency analysis diagrams of microseismic signals, multiple overlapping seismic sources within different frequency ranges can be distinguished, and the contributions of different river processes can be extracted, thus enabling real-time monitoring of various river processes.

### 3.1 Human noise

Human activities can also cause seismic disturbances, such as industrial activities, vehicle noise, or people walking near seismometers. Although noise from human activity is highly variable, some typical time-frequency characteristics of noise from human activity can be highlighted. The river section studied in this research is located next to a road. During the monitoring period, vehicles driving on the road may generate signals that could affect the experiment, which is the main source of human activity during this research period. To exclude the interference from human activity, we placed the base station S3 1.5 meters away from the road in four experiments to monitor the human activities noise during the experimental period. Vehicles passing through the research section represent individual disturbances, which are of short duration and mainly affect the frequency range from 2 to 40 Hz, with high signal amplitudes and rapid attenuation of high-frequency signal amplitudes. The signal is more susceptible to attenuation with increasing distance, and is only recorded at nearby stations. As can be seen from the waveform of the microseismic signal at the S3 base station in the second experiment, the microseismic signals generated by passing vehicles are rarely coherent over the entire monitoring array (Figure 6). Therefore, the noise from human activities is not the main source of seismic energy in this study and can be largely filtered



out with a bandpass filter.

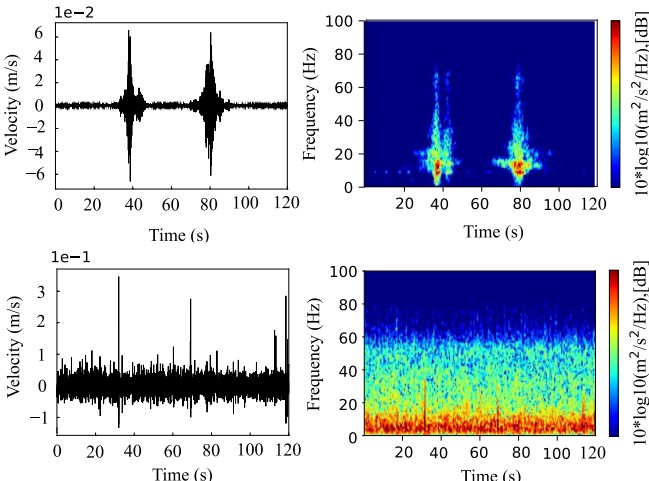

**Figure 6. Waveforms and spectrograms of microseismic signals generated by the river and vehicle travelling**
**in test2. Top: seismic signal emitted by vehicle recorded by S3 in test 2; bottom: seismic signal emitted by**
**river recorded S2 by in test 2, 20 meters away from S3.**

## 3.2 Still water flow

Except for the river section in the fourth experiment, which is located in a position with a steep slope and
high water flow rate, the water flow in Test 1, Test 2 and Test 3, was relatively slow. Analyzing the
microseismic signals and their spectral characteristics produced by the rivers in Test 1, Test 2 and Test 3,
the seismic responses recorded by the monitoring stations in the three river sections share many
similarities. Throughout the entire monitoring period, the seismic signals exhibit a clear broadband
(2~50Hz) seismic response in both horizontal and vertical components. From the time-frequency analysis
and spectral plots of Test 1 (Figure 7), the energy of the microseismic signals is distributed across the
1~50 Hz frequency band, with most of the energy concentrated in the 2~16 Hz band. In Test 2, the energy
of the microseismic signals is distributed across the 1~60 Hz frequency band, with most of the energy
concentrated in the 2~12 Hz band. In Test 3, the energy of the microseismic signals is distributed across
the 1~50 Hz frequency band, with most of the energy concentrated in the 2~10 Hz band. By performing
time-frequency analysis on the microseismic signals recorded by the monitoring stations closest to the
river channel in the three experiments (namely S4, S2, and S4 stations) and calculating their Power
Spectral Density (PSD), the results plotted in the same graph show that for these three experiments the



energy distribution is mostly concentrated in the 2~15 Hz frequency range (Figure 8).

Previous studies showed that river flow and its variations tend to excite low-frequency seismic power (1-
10 Hz), that there is a significant correlation between anomalous microseismic signals in the 2-10 Hz
band and river flow variations, and that the seismic power at 50 Hz has a linear relationship with the
measured sediment fluxes in the riverbed (Burtin et al., 2011; Gimbert et al., 2014; Tasi et al., 2012; Díaz
et al., 2014). The main frequency bands observed in this study are generally consistent with the
conclusions drawn by previous scholars, and there was no significant sediment transport process during
the experimental period. The seismic energy is mainly contributed by turbulence, so we can infer that the
low-frequency band of 2~10 Hz in the experimental spectrum is related to the turbulent flow process of
the river, and the changes in its energy reflect the changes in river flow rate.

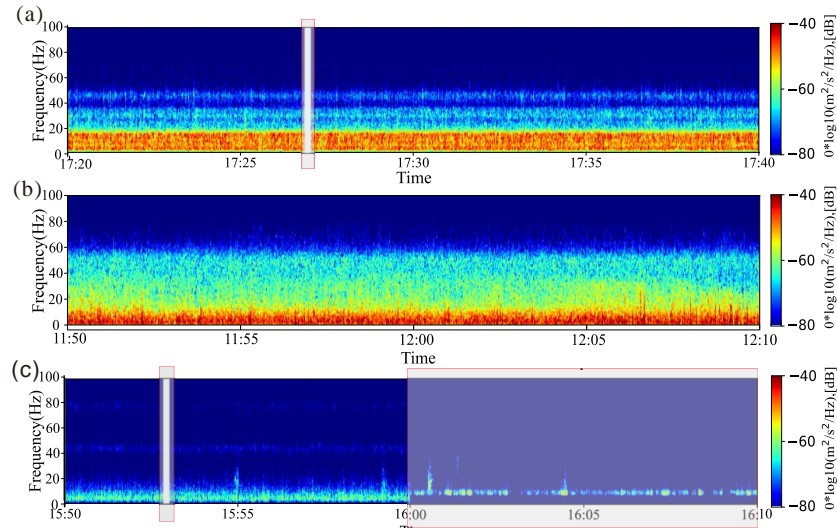


**Figure 7. Spectrograms of the microseismic signals generated by the river. a), b), c) River spectrograms of the**
**microseismic signals generated by the river at locations 1, 2 and 3 of the experiment, respectively (missing**
**data due to instrumental interruptions are in the red boxes).**



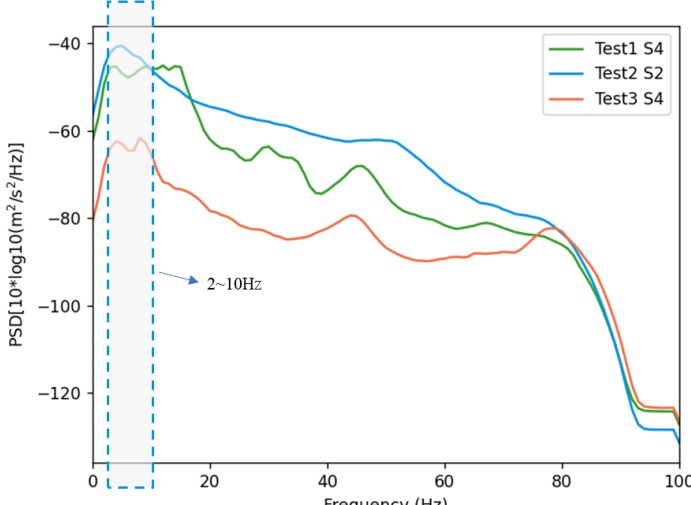


**Figure 8. Acceleration power spectral density plots of microseismic signals generated by rivers. The green**
**curve represents the PSD curve of the water flow at the nearest base station S4 of the river in test 1, the blue**
**curve at base station S2 in test 2, the orange curve at the base station S4 in test 3; the blue dashed box**
**highlights the frequency band of maximum energy distributions of microseismic signals of tests 1, 2, 3.**


**3.3 Turbulent river and sediment transport**
Geophones can detect elastic waves generated by processes occurring at or near the Earth's surface. These
elastic waves result from the transfer of energy produced by objects striking the ground. The sources of
elastic waves generated by river processes can be quite complex, depending on the flow configuration of
the river. These sources include particle collisions during sediment transport, water turbulence, bubble
explosions, and the propagation of gravity waves or breaking waves on the river's surface (Figure 9).
Sediment transport encompasses various particle movements, such as suspension, rolling, hopping, and
sliding (Boano et al.,2011).
These river processes induce vibrations in the riverbed, generating elastic waves that propagate through
the ground medium as vibrational signals. Within the range where the signal energy dissipates
completely, the deployed microseismic stations can receive these signals and record them as



corresponding voltage fluctuations.

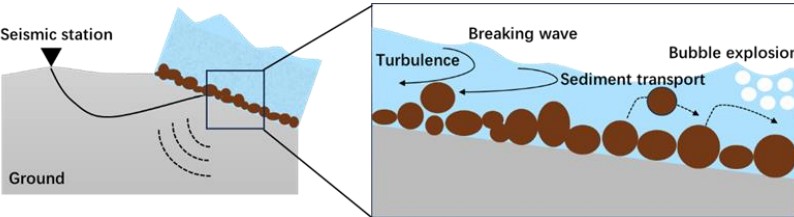


**Figure 9. Seismic noise generation by turbulent flow in rivers. The brown ovals represent gravel particles in**
**the river, which generate microseismic signals as they move with the current, the white ovals represent**
**microseismic signals generated by the explosion of air bubbles in the water.**


During the research period, the fourth experiment was conducted in a section of the river channel with
large boulders that create a certain drop in the riverbed. The riverbed in this area is composed of gravel
and pebbles, with the largest gravel size measured at 50×36×20 cm. The time-frequency analysis and
spectral plots from this location indicate that the energy of the microseismic signals is concentrated in
two distinct frequency bands, namely 2~15 Hz and 35~50 Hz, with the maximum energy located in the
7~15 Hz band (Figure 10a). The time-frequency plot from Test 2 shows that in sections of the river
channel where the gradient is gentler, the energy of the microseismic signals is primarily concentrated in
the 2~12 Hz frequency band (Figure 10b). Research by Burtin et al. (2011) indicates that river flow and
its variations tend to excite low-frequency seismic power (1~10 Hz). Schmandt et al. (2013) found that
between 35~50 Hz, this band includes frequencies (15~45 Hz) previously identified as being excited by
sediment transport in river studies. Based on the data from this experimental study, it can be inferred that
the seismic energy in the fourth experiment mainly originates from the river and the flow of river water
driving a small amount of sediment transport to produce microseismic signals. The low-frequency band
of 2~15 Hz in the spectral plot is related to the river flow process, while the higher frequency band of
35~50 Hz is associated with the river's impact on the riverbed, a small amount of sediment transport, and
human activities (Barrière et al., 2015; Bagnold et al., 1966; Turowski et al., 2016). Since the river studied
in this research is a low-flow river system, with no significant sediment transport phenomena observed
during the experiment, a detailed exploration of the characteristics of microseismic signals generated by
sediment transport was not carried out.



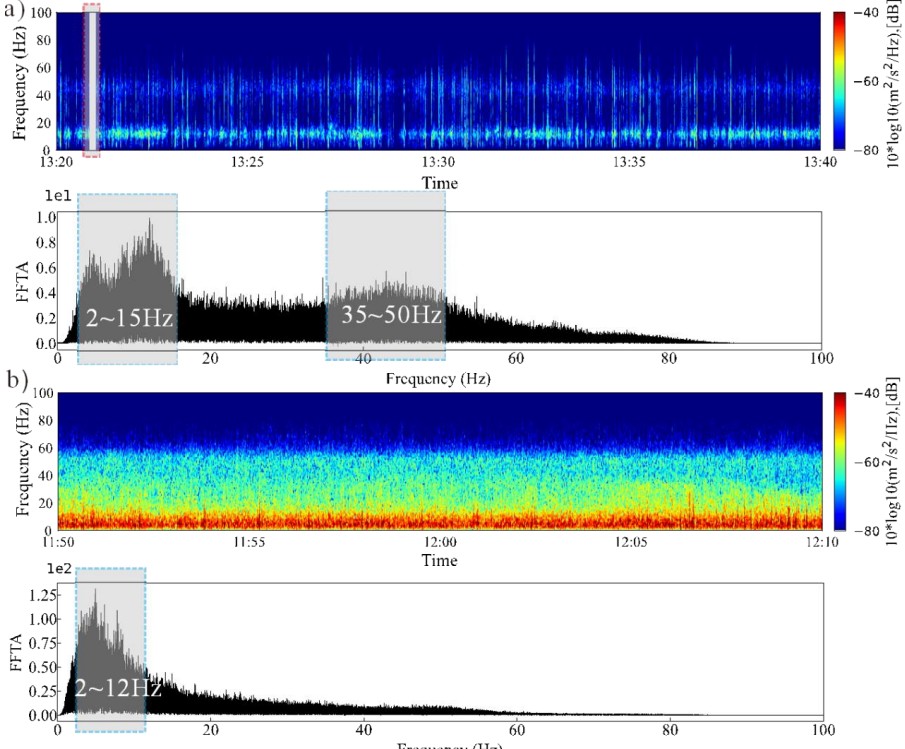

**Figure 10. Time-frequency diagrams of microseismic signals generated by rivers and their spectral characteristics. a) is the time-frequency analysis and spectral characteristics of the river with a certain drop of the river section at four places of the experiment; from the time-frequency analysis, it can be seen that the energy of the microseismic signal is concentrated in the more obvious two frequency bands, respectively, 2~15 Hz, 35~50 Hz; b) is the time-frequency analysis and spectral characteristics of the river at two places of the experiment in the gently sloping section of the river, and the microseismic signal is mainly concentrated in the 2~12 Hz frequency band. The red dashed box shows missing data caused by the instrumental interruption, and the blue dashed box shows the frequency band where the energy of microseismic signals is mainly distributed.**

## 4 Seismic interpretation and river discharge calculation

### 4.1 Seismic data processing

Geophones receive signals generated by rivers and record them as corresponding voltage fluctuations. These are then converted back into ground velocity based on the characteristics of the microseismic instruments, allowing for the creation of the most primitive form of seismic waveforms produced by the river signals, that is, the waveform or time series of ground velocity. The seismic amplitude of the time





series can provide information about the seismic signal, but some important features may be obscured
by input unrelated to the monitored event. Filter processing for specific frequencies can help to filter the
relevant parts of the signal. Transforming the signal to obtain the spectrum is a powerful tool for
quantifying the signal amplitude in the frequency domain. It allows for rapid characterization of the
signal and can be used with specific frequency filters to test and filter specific signals. However, in the
spectrum, the time information is not decomposed, and the fluctuations of the spectrum in the time series
are unknown. A common tool for characterizing seismic signals is time-frequency analysis, which
combines both aspects, allowing the amplitude or energy of microseismic signals to be quantified in both
the time domain and the frequency domain.

The Fourier transform is a classic method of time-frequency analysis. Using the Fast Fourier Transform
(FFT), continuous microseismic signals are divided into short segments, and a taper is applied to the
segments to obtain the spectral plot of the microseismic signal, thereby showing the distribution of
seismic energy in both time and frequency. To reduce the spectral variance typically caused by the simple
use of FFT and to quantify the energy produced by microseismic signals at a given frequency, we
calculated the Power Spectral Density (PSD) of the time series using Welch's overlapped segment method
(Welch,1967). The time series is divided into several overlapping segments, and to avoid errors when the
signal is truncated, a Hamming window is used to window the segments, with a 1-second (200-sample)
window having a 50% overlap, thereby obtaining a discrete 1Hz frequency band.

**4.2 The relationship between river discharge and seismic noise**
By discussing the source of seismic waves in the fourth section, it can be concluded that the seismic
energy of the river in this study is mainly due to turbulence. The low-frequency band of 2~10 Hz in the
experimental spectrum is related to the turbulent process of the river, and the changes in seismic energy
reflect the changes in river flow. For verification, we selected data from the third experiment. During the
third experiment, while monitoring the microseismic activity of the river, we simultaneously conducted
continuous measurements of river flow velocity, measuring the average flow velocity at a distance of
1.35 m from the riverbank every minute. Based on the flow calculation formula in Section 2.3, we
obtained the average flow rate per minute. The microseismic signals were selected from the S4 station,



which had the highest signal-to-noise ratio. Using the method of estimating the average power spectral
density of the signal, we calculated the average seismic power at each frequency over a 1-minute period,
and then calculated the average seismic power of the microseismic signal in the 2~10 Hz band over a 1-
minute period. This was converted into energy form and plotted on the same time axis as the flow rate
changes (Figure 11). It can be clearly seen from the figure that the fluctuations in the average seismic
power of the three components in the 2~10 Hz band recorded by the S4 station basically match the
fluctuations in the river flow at the same time, and the two have good consistency on the time scale.
There are differences in the average seismic power of different components, but there are no obvious
differences in the response to changes in river flow. Among them, the average seismic power change of
channel N is highly consistent with the change in river flow and can well reflect the fluctuations in river
flow. The above results indicate that there is a strong correlation between the recorded microseismic
signals in the 2~10 Hz band and river flow. Real-time monitoring of river processes can be achieved
through the analysis of the time-frequency diagram of continuous microseismic signals.

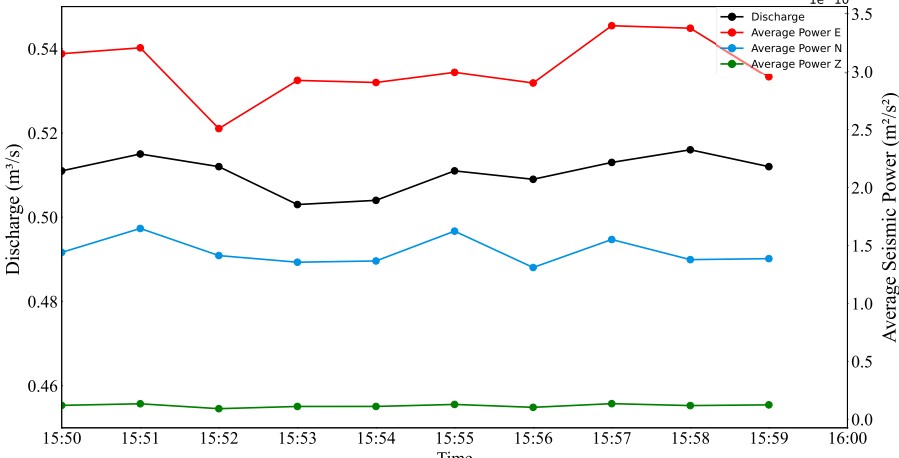


**Figure 11. Mean seismic power and mean river flow correlation for Test 3. The black line represents the**
**average flow, the red, blue and green lines represent the average seismic power of the geophone in the east-**
**west (E), north-south (N), and vertical (Z) directions, respectively.**


**5 Results and discussion**
This study employs a linear least squares regression model to quantify the relationship between the
seismic power spectral density (PSD) above the 1 Hz band and the river turbulence, without considering



the mechanical effects generated by the river process. The first 10 minutes of microseismic data from
each location during the experiment were used for regression; these regressions were then used to
calibrate the least squares model for flow prediction. The last 10 minutes of microseismic data from each
location during the experiment were then used for testing, ultimately resulting in a linear approximation
model for inferring river flow from microseismic data; the flow of the river section in the third experiment
was not inverted this time, as the microseismic data for this time window experienced anomalies due to
instrument issues, and could not be used for flow inversion. The total energy of the seismic waveforms
generated by multiple sources is the sum of the energies in the river processes of each river section.
Without considering the seismic energy generated by river sediment transport, the total seismic PSD is
the sum of the PSDs produced by water turbulence ($P_Q$), passing vehicles ($P_V$), and environmental noise
($P_N$). When constructing the linear least squares regression model, it is assumed that the PSD generated
by the first-order river turbulence process is linearly scaled with the variable representing the magnitude
of that process. Therefore, for any given time t, using the seismic PSD at each frequency band ($P_f$), the
PSD generated by road vehicles ($P_{vf}$), the average power of station S3 at each frequency during periods
without human noise such as vehicles ($P_{Nf}$), and the constant linear coefficient af of flow at frequency $f$,
the prediction equation for river flow $Q_{pred}$(t) at time t is obtained:
$$Q_{pred}(t) = [P_f(t) - P_{vf}(t) - N_f]/a_f \qquad\qquad (8)$$
The above equation provides the flow prediction for each frequency, which is solved using the least
squares method to maximize the consistency of predictions across the 1~100 Hz. The flow regression
coefficients corresponding to the highest turbulence signal-to-noise ratio geophones for each component
(E, N, and Z) and each experiment site (Test 1, 2, and 4), calculated using the first 10 minutes of
microseismic data, are shown in Figure 12. The regression coefficient $af$ encompasses both the coupling
between the flow measurement unit (m$^3$/s) and the ground velocity signal produced by turbulence, as
well as the attenuation of each microseismic signal between the source and the geophones (the Green's
function). This coefficient also represents the spectral contribution of the turbulence process to the
geophones signal, or the power per frequency transmitted by unit flow.





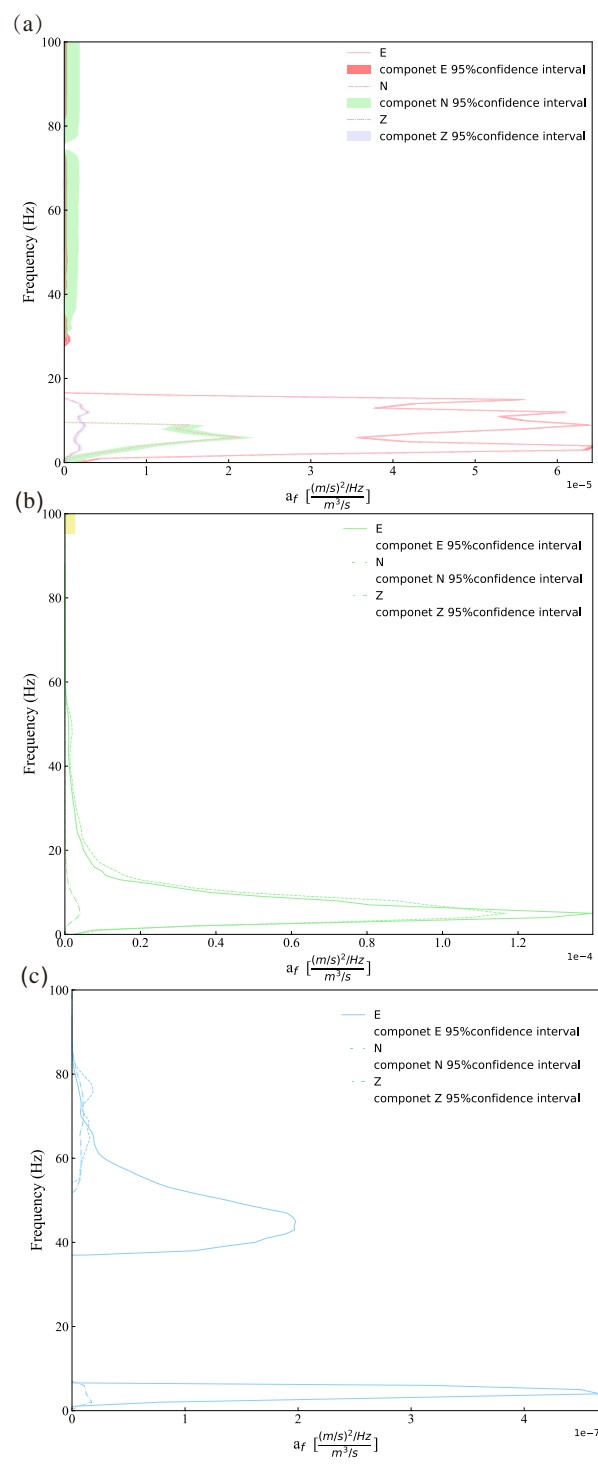

**Figure 12. The flow regression coefficients and 95% confidence intervals for the river processes. (a), (b), and**



**(c) show the flow regression coefficients and 95% confidence intervals for the river processes on the E, N, and**
**Z components of the ground motion at base stations S4, S2, and S4 at test 1, 2, and 4, respectively.**


After regressing the constant linear coefficient $af$ of flow at frequency $f$ using the first 10 minutes of
microseismic test data, the microseismic test data from the last 10 minutes is used to calculate the
predicted flow values at each experimental site using the aforementioned equation. Since the flow rate
calculation in this study is derived from the measured 1 minute average flow velocity, to ensure that the
predicted results match the measured data, the peak values of the flow-related turbulence coefficients $af$
(at frequencies of 4 Hz, 5 Hz, and 4 Hz) for the first, second, and fourth tests are substituted into the
equation to calculate the river flow predicted values at every 1 minute interval during the last 10 minutes.
The predicted results are compared with the flow calculation results from Section 2.3, as shown in Figure
12. From the comparison chart of the two, it can be seen that the predicted flow values established by the
linear approximation model in this paper are close to the actual observed values, with the predicted flow
values fluctuating around the corresponding observed values. The average values of the predicted flow
values for the first, second, and fourth tests are 0.454 $m^3/s$, 0.548 $m^3/s$, and 0.537 $m^3/s$, respectively. The
average values of the predicted flow values indicate that the results predicted by the model in this study
are relatively accurate. However, the average absolute errors between the predicted flow values and the
measured values for the three tests are 0.030, 0.080, and 0.237, respectively. Moreover, from Figure 13,
it can be visually observed that there is a larger fluctuation between the predicted flow values and the
measured values for each minute of the fourth test. This may be due to the presence of large boulders in
the river section of the fourth test, which create a drop in water flow and impact the riverbed, generating
noise caused by sediment transport. Additionally, during the fourth test, there were trucks and excavators
operating about 150 meters away from the river section, which may have caused larger fluctuations in
the predicted flow values for each minute of the fourth test. During the entire data preprocessing stage,
all microseismic data were high-pass filtered to exclude the noise influence below 1 Hz, to reduce as
much as possible environmental noise produced by vehicles, construction works, and other human
activities,. Therefore, in this test, there is still an error between the flow prediction results obtained by
inversion and the actual test results also due to the impact of the instruments' installation point, to the
river flow calculation method, and the instrumental errors itself. In the future, more accurate experimental
results could be obtained by starting from traditional flow refinement calculations and precise filtering





of river microseismic signals, increasing the number of microseismic stations, and building on the
proposed model to further refine the inversion analysis.

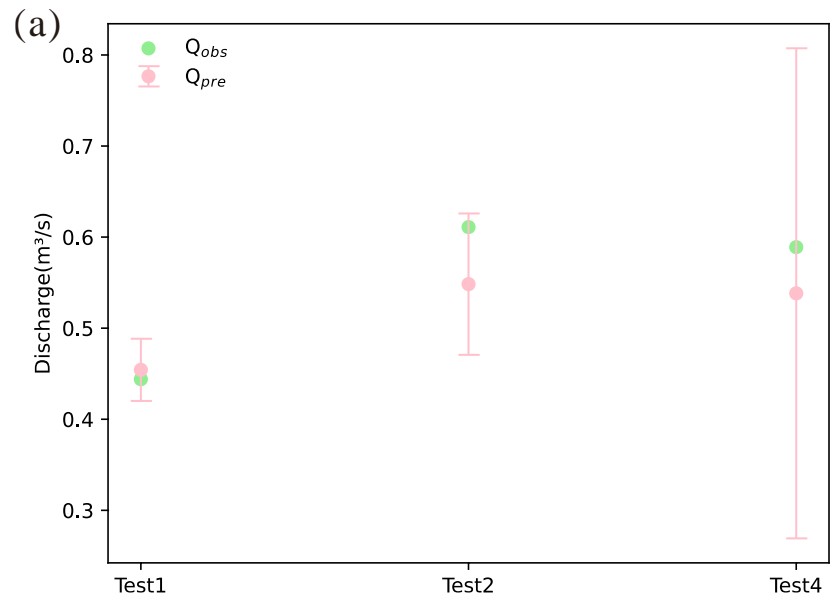


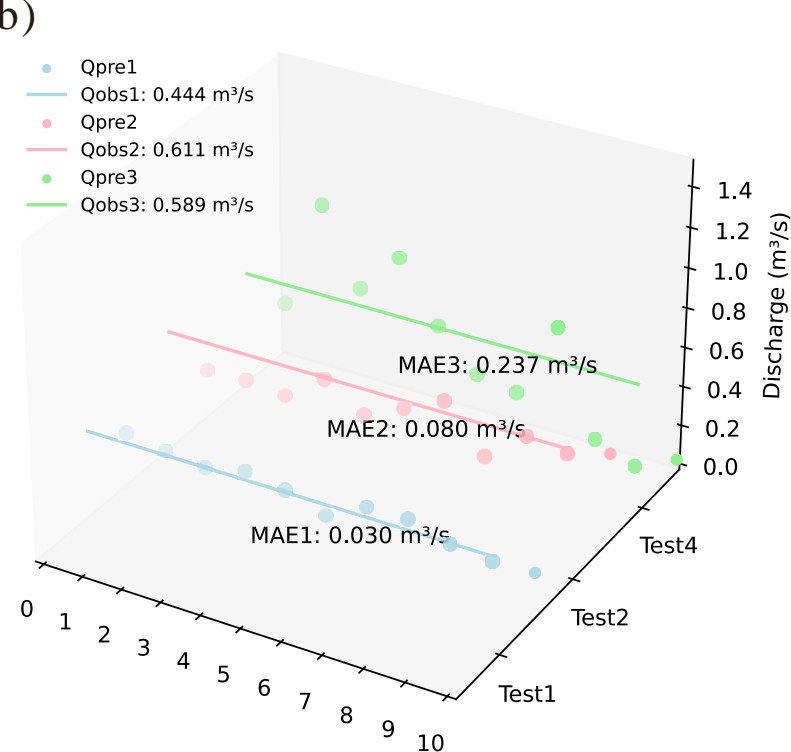




**Figure 13. The plot of measured flow values against inverted flow predictions, (a). The scatter plot of mean absolute error between flow predictions and measured values, (b). In figure (a), green represents measured values and pink represents predicted values. In figure (b), blue represents Test1, pink represents Test2, green represents Test4, straight line represents measured flow values and scatter represents predicted values.**

The linear approximation model proposed in this study for flow inversion has a relative error within 10.3%, but it also has some limitations. First, in practical situations, there is a slight nonlinear relationship between flow and seismic power, which the established model does not consider. This omission can affect the accuracy of the model inversion, leading to a reduction in inversion accuracy during practical application. Additionally, the linear approximation model proposed in this paper is built for the specific river environment in question and can be generalized to similar river environments. However, for rivers in different environments, other factors contributing to seismic power need to be considered.

**6 Conclusion**

The study analyzes the seismic records from 3 to 4 three-component seismometers deployed across four sections of a low-flow river system in the village and combines measurements of flow velocity and cross-sectional area of the river sections to calculate its flow data. We found that the signals generated by the river flow have a very wide frequency range (2~50Hz). Despite the presence of noise fields generated by human activities throughout the research process, which are mainly high-frequency acceleration energy, we cannot establish a correlation between high-frequency seismic power and river flow. In contrast, the recorded microseismic signals in the 2~10 Hz band have a strong connection with river flow, approximately exhibiting a linear relationship. Moreover, the microseismic signals generated by turbulence have frequencies lower than those produced by human activity noise and riverbed sediment transport and can be separated using a bandpass filter. Even with low river flow, real-time monitoring of the turbulence process can be achieved through the analysis of continuous microseismic signals time-frequency diagrams.

A linear least squares regression model was used to quantify the relationship between seismic power spectral density (PSD) above the 1 Hz band and the river turbulence process, without considering the mechanical effects generated by the river process. The first 10 minutes of microseismic data from each



location during the test were used for regression, and these regressions were then used to calibrate the
least squares model for flow prediction. The last 10 minutes of microseismic data from each location
during the experiment were then used for testing, ultimately resulting in a linear approximation model
for inferring river flow from microseismic data. The predicted average values obtained from the inversion
experiments for Tests 1, 2, and 4 are 0.454 $m^3$/s, 0.548 $m^3$/s, and 0.537 $m^3$/s, respectively, with the
maximum relative error between the predicted and measured values being 10.3%. By analyzing the
microseismic signals generated by vehicles and recorded at S3 in the third test and their spectral
characterization, the microseismic signals in the 2~7 Hz band were retained by using a band-pass filter
to minimize the impact of vehicle-related signals. The 1 minute average seismic power in the 2~7 Hz
band on the time-frequency analysis map of the recorded microseismic signals was calculated by
estimating the average power spectral density (Welch) of the signals and converting it into energy form;
the calculated results were in good agreement with the 1 minute average flow rate measured in the field
in the time scale, and showed that the time-frequency analysis based on continuous microseismic
monitoring of the river can enable the monitoring of the river processes.
**Author contributions.** ZXY conducted the experiment, analyzed the experimental data and wrote the
manuscript. ZSZ and XW participated in the experiment. WYM, XB and Emanuele Intrieri provided
revision suggestions for the manuscript. FL guided the implementation of the experiment and revised the
content of the manuscript.
**Competing interests.** The authors declare no competing interests.

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
