# Peer review of "A physical model for mean river discharge calculation"

_EGUsphere, 2024_

## Referee Comment (RC1)

**Review for egusphere-2024-4111, title: A physical model for mean river discharge calculation: from riverside seismic monitoring experiments in a low flow river, China**

The manuscript by Xiaoyue Zhou and co-authors investigates the relationship between flow discharge and seismic data along a natural reach of the Jiuqu River, a tributary of the Meishui River in China. Using seismic geophones alongside flow depth and velocity measurements, they link flow phenomena to seismic signals generated by forces acting on the riverbed during a one-day monitoring period. Their analysis distinguishes between noise from human activity and seismic signals related to stream flow, identifying turbulence as the primary contributor to seismic waves at frequencies below 10 Hz. They test whether seismic power can be used to estimate stream discharge (Figure 11) and propose an empirical equation that incorporates contributions from various environmental sources. By calibrating their model on part of the data, they determine a coefficient that allows them to predict flow using unseen data. Their model performs well across three experiments, aligning closely with discharge measurements (Figure 13). The study underscores the potential of continuous microseismic monitoring for quantifying river processes and isolating turbulence signals, even in low-flow conditions.

While the manuscript shows potential for enhancing discharge predictions using seismic data, it currently lacks several key components that must be thoroughly addressed before it can be considered for further evaluation and publication. Below, I outline my main concerns, followed by specific comments and line references from the main text that can guide the authors in revisiting the manuscript.

1. The manuscript, in its current form, lacks a clear motivation. Throughout the text, the authors' objectives are not sufficiently articulated. What is the main purpose of the project? Why were the experiments conducted? What was the rationale behind the choice of station locations? Additionally, why was the monitoring period limited to just one day? Please also refer to the last paragraph, after point (5).

2. The structure of the paper is not well organized, which distracts the reading process. This may be partly due to an insufficient definition of the paper's scope (see my above point 1). The Introduction section lacks clear organization, with discussions of seismic methods interspersed throughout and paragraphs lacking coherence. Additionally, the focus on the actual flow phenomena is insufficient and should be given more emphasis (see my comments on the first paragraph of the introduction). Moreover, many methodological details are only introduced in the Results and Discussion sections. I recommend a thorough reorganization of the manuscript, beginning with an introduction to flow phenomena and possibly a brief discussion on sediment transport. The authors can then identify research gaps and explain how environmental seismology can address these gaps. The research gaps must be then connected to the authors' current work.

3. The figures are not evenly distributed throughout the manuscript and require revision. For example, the first three figures could be combined into a single, more cohesive figure. They should also better support the main text by helping the reader understand the rationale

behind the seismic experiments. Additionally, schematic representations should be replaced with aerial or field photos (see my specific comments below).

4. The empirical model used by the authors (Equation 8) closely resembles the one proposed by Roth et al. (2016), and this should be acknowledged. Additionally, the main title ("A physical model for...") does not align with the empirical nature of Equation 8. The authors should either provide a justification for why this model can be considered physical, or revise the title to more accurately reflect the empirical nature of the model and the main message of the paper.

5. The justification for the hydrological perspective needs to be strengthened. The authors introduce the field settings and describe the low-discharge regime during the survey period. They then calibrate their empirical model (Eq. 8) and apply it for discharge predictions. However, they should explicitly present the range of discharge values measured (or estimated using their approach) on the survey day. Specifically, what was the observed discharge range during the seismic and hydrological survey? If the fluctuations remain within a narrow range (e.g., within a factor of 1, meaning no substantial deviation from the average), then applying the model within the same range, even to unseen data, may have limited value. To address this, the authors should first present their discharge time series and then assess whether their model is applicable over that range. This clarification is crucial for strengthening the manuscript's narrative and overall scientific rigor.

In light of the above points, I believe the manuscript has not yet reached a stage where a brief revision would suffice. Major or thorough revisions are necessary, particularly due to the lack of a coherent key message and a clear demonstration of the uniqueness of the approach. Specifically, and this is a critical point of my review, the manuscript needs to clarify how this work advances our understanding of monitoring flow phenomena using seismic tools, especially in comparison to previous studies (e.g., Gimbert et al., 2014; Schmandt et al., 2013; Lagarde et al., 2021; Roth et al., 2016). Once the key message is clearly established, the manuscript should be reorganized for better clarity, to more effectively present the arguments, objectives, and overall scope.

**Specific comments**

**Introduction**

I believe the Introduction section needs better organization. The current presentation of the motivation for flow monitoring - specifically, flow velocity and bedload transport - is unclear, and the rationale for using seismic monitoring is missing.

I suggest restructuring the introduction, e.g., as follows:

1. First paragraph: Introduce the key phenomena of river flow and bedload transport, emphasizing their importance and the need for improved understanding.

2. Second paragraph: Introduce environmental seismology as a remote sensing technique that mitigates risks associated with direct measurements of these processes. Briefly mention the key studies published in the realm of fluvial seismology, categorizing them by their focus (e.g., studies on flow, studies on bedload transport).

3. Third paragraph: Use specific examples (e.g., Schmandt et al., 2013; Burtin et al., 2011; Roth et al., 2016) to provide a foundation for later discussions in the manuscript.

4. Fourth paragraph: Identify the key research gaps that remain in the field and explain how the current study contributes to addressing these gaps. Emphasize how the manuscript provides a scientific evaluation of the proposed approach and its implications for advancing the understanding of river dynamics.

This restructuring will improve the logical flow and strengthen the manuscript's narrative.

Lines 33-35: I suggest to include an opening sentence which can then be used to argue for the importance of river monitoring.

Lines 34-35: there is also an empirical approach, where equations are used (e.g., for bedload transport and flow velocity).

Lines 38-39: For the purpose of your argument, I would change the references of Roth, Cook and Larose, since these are directly Environmental Seismology studies, while your opening paragraph discusses river monitoring.

Lines 40-42: The introduction of the seismic methods comes without a scope and I think you should first establish the reasoning for the use of seismic signals to infer river activity.

Lines 43-36: Sentence is redundant, and I don't see how it contributes to your arguments in this paragraph. Either reformulate or emit.

Line 46: "microseismic" – since this term repeats in your manuscript, and since it is used differently in most (if not all) the environmental seismology-related papers, I propose to either

change it to "seismic signals" or to define what is "microseismic signal" (as opposed to a "seismic signal").

Line 49: "water surface".

Line 52: what do you refer to by using "This"? be specific.

I also find that your reasoning here lacks a strong argument, that is, "This demonstrated that … can not only aid in studying….", but the above sentence is not convincing enough. I propose to revise the sentences between lines 46 and 52 to better present the arguments (using the studies you mentioned – Schmandt and Diaz) leading to the motivation of using seismic instruments to understand river flow.

Lines 57-59: A meaningless sentence. If the purpose of this paragraph is to elucidate the different seismic signatures obtained by various authors, maybe you should begin with a different sentence, emphasizing the main features of the seismic signal (frequency, Power Spectral Density – PSD).

Lines 57-69: Since you are using previous references that using seismic signals to demonstrate it can be applied as a passive tool to infer flow and bedload phenomena, it would be worthwhile mentioning both field-based (e.g., Schmandt et al., 2013; Roth et al., 2014; Burtin et al., 2011, and several more) as opposed to theory-based  (i.e., Tsai et al., 20212; Gimbert et al., 2014; Luong et al., 2024) studies.

Lines 71-78: The entire paragraph reads detached from the rest. You begin with "warning systems…", then you go on discussing the general method of analysis ("by analyzing the time-frequency…"). but it feels a more general description of the method, so it needs to come earlier, where you introduce environmental seismology.

Line 80: "This study focuses on the monitoring…" – I am lacking a motivation description to your study. This should come in the above paragraph. In what way does your experiment adds on previous experiments that conducted field experiments with flow and seismic measurements? Is it related to the flow regime in your study sites? Is it related to the number of your monitoring sites? Explain.

In addition, in this stage of the intro you should have already laid out the objective to your study, what were your aims? It is not clear by reading this paragraph what were you aiming to do.

**2 Experiments**

Line 89: I propose to change the title from "Experiments" to "Methods".

Line 91: "We studied the Jiuqu River, a tributary of…"

Lines 92-93: Instead of refereeing to "Meishui town", refer to either the river or the catchment itself.

Line 93: what is a relative height? Do you mean Relief?

Lines 94-95: Is the description of lithology important to your scientific work presented here? If it does not serve an important information ,I propose to remove it.

Line 97: either "four field monitoring experiments", or "four field experiments"

Line 98: what are current meters?

How do these measure flow velocity of the flow when situated o the banks?

Line 99: "for this experiment", what do you mean by "this?"

Line 102: I am unclear what is a drainage canal.

Line 104: when describing grain size, please refer to standard jargon, for example, using its axis (axis a, b and c).

Line 105: I think you would benefit from reporting the averaged length of axis b itself, rather than the volume.

Line 106: Was your sediment concentration meant for the purpose of evaluating sediments travelling in suspension? Clarify?

From what elevation above the bed were these measurements conducted?

Line 108: "which classifies it as a low-flow river" – according to what standards? Please give reference, or revise\emit.

I am unclear how the reference to Figure 2 in line 108 established the argument for a low flow.

Line 119: "seismic ambient noise was collected" – the noise was not collected. Measurements were taken, or seismic noise was monitored. Please revise.

Lines 119:121: "Seismic instruments offer a variety of…" – delete this sentence.

Line 122: why is it important to mentions what instruments are categorized? You are not reviewing seismic methods. You use a relatively well established method, so use a reference for that.

In general, I find that the paragraph starting in line 119 and ending at 132 can be reduced to one simple, concise  sentence.

Line 134: not seismic stations, but seismic instruments.

Line 139-140: I do not understand the sentence (one of which was an integrated …"). Revise or emit.

Paragraph encompassing lines 134 – 145 is too informative. Reduce and leave only the most important information.

Line 150: "Match 17" – the monitoring period was for one day? What was your purpose in conducting an experiment for one day? How long within that day did your experiment last?

Lines 150-151: move this to the beginning of the Method section to emphasize your objectives.

Line 158: what is the second experiment? I find it a little bit challenging to find my way within the introduction of your method section structure. For example, when you write "For the second experiment…" would imply for a reader that there are a few experiments. In such a case, you need to introduce them properly, before you delve into explaining them in detail.

Line 160: "Specifically, S2 was…" – it make a little sense to explain about S2 when you did not introduce it. Begin with a proper introduction of the experiments, and the stations.

Line 163: Why did you use four stations? In Figure 3, it seems that the stations are very close to each other. What were your aims in deploying for, and not only one stations? This needs to be justified.

Lines 163 – 169: I find this paragraph more suitable for the Results section.

Line 180: "calculating river flow" – what is river flow? Please specify, is that discharge (e.g., in units of $m^3/s$) or is it flow velocity (e.g., in units of m/s)?

Line 186: again, I think you mean here "discharge" rather than "river flow".

Line 193: the title feels detached. Revise to make clear what you are doing.

Lines 213-214: please fix the equation so that it is presented near its number (7).

Line 218: "tests" – do you mean experiments? Please be consistent with the terms you use.

**3 Seismic ambient noise**

Line 220: change title to "Results". It was not clear from the title that this is the results section, and I was reading a substantial part of this section believing it is still the methods (see comments below).

Lines 221-222: This comment applies specifically to the mentioned sentence but can be generalized to similar instances throughout the manuscript. It is crucial to maintain a clear distinction between the Introduction, where you present the state of the science, previous studies, and relevant fields, and the Methods, where you concisely describe the techniques used, ensuring they are theoretically reproducible. For example, in this case, rather than directly explaining the seismic ambient noise method, lines 221–231 primarily serve to justify the approach. However, such rationale belongs in the Introduction rather than the Methods section. Ensuring this distinction throughout the manuscript will improve clarity and logical structure.

Line 239: please specify the distance of the road from the experimental sites.

**Figures 1, 2 and 3:**

I believe the manuscript would benefit from a more integrated and visually informative representation of the experimental sites. Instead of separate schematic cross-sections and contour-like maps, I suggest merging Figures 1, 2, and 3 into a single figure with multiple panels. This would provide a more comprehensive overview of the study area by: (1) including a small inset map to show the study location within a broader China map for georaphic context, (2) using an aerial or satellite image to display the experimental sites along the river, (3) incorporating photos of the different study sites to highlight their distinct morphological characteristics, and (4) ensuring a clearer depiction of station locations within the field setting, as the current photo in Figure 3 does not make their placement evident.

Additionally, the sensors in Figure 3 appear wired to cables - are these temporary? More details on the station setup, location, and monitoring period would improve clarity. Lastly, the red sensor between S2 and S4 lacks a label— - please clarify its designation. This restructuring will improve the figure's effectiveness and enhance the manuscript's clarity.

**Figure 5:**

In the figure's caption, please explain all the variables you mention in the figure.

**Figure 6:**

I believe this figure is redundant in the main text as it is secondary to your main analysis. Therefore, I propose to move it to the supplementary information.

Line 255: What is "still water flow"?

Lines 256-257: please specify "experiment" and "test". Are there differences between these terms? Are these the same?

Lines 257-260: This sentence is redundant. Delete.

Line 260: "A clear broadband seismic response" – in what way it is clear? That is subjective. I would try to reformulate the sentence to a more objective one.

Lines 272-280: This entire paragraph in an interpretation, not a result. Please move it to the discussion if needed.

**Figure 8:**

"Acceleration power spectral" – why do you use acceleration rather than velocity?

"PSD plots" – I think these plots would be better called "seismic spectra".

Why do you choose to exhibit three curves and not all of the curves that you have as data sets?

Lines 293-299: Again, this paragraph belongs to the introduction. Remember that you are now introducing your results, so at this stage the reader should already be informative of the fact that "geophones can detect elastic waves generated by processes…". In fact, you already wrote this previously, so it is a repetition. In this manner, the allocation of Figure (9) is not well suited

Lines 300-302: You are mixing pure results (e.g., the measurement showed that…) with discussion, which is meant to bring forth your interpretation of the results (e.g., "the deployed microseismic stations can receive these signals and record them as").

Lines 310-312: You introduce the fourth experiment, but this should already have been done previously in the methods section.

**4 Seismic interpretation and river discharge calculation**

Line 340: please change this title to "Discussion.

Lines 342-354: this entire paragraph also reads detached from the text. As this is the beginning of the discussion, you should more pronounced focus on bringing forth your results and discussing them with respect to previously published science in this field.

Lines 356-364: Same comment as above. This paragraph should be deleted. The information is interesting, but it is not related with your specific work. The focus should be on interpreting the results, not the methods, which are probably massively discussed elsewhere in the literature.

Lines 367-368: The sentence is superfluous.

Lines 368-370: It is too early to state that. You should begin with you aim, proceed with what you did as a test (as you begin in line 370, "for verification"), then conclude with something in the form of "as a result, we propose that the frequency range within 2 and 10 Hz records seismic signals generated form turbulence…".

Alternatively, you could write something along the lines of: "We hypothesize that the low-frequency… in the experimental … is related to flow turbulence within the stream". To validate our hypothesis, we plot…".

**Figure 11:**

This figure should be within the results, not discussion. The discussion n can pick this up by discussing in what ways does this plot reinforces (or not) your hypothesis that turbulence dominates the < 10 Hz frequency band.

**5 Results and discussion**

Line 394: it does not make much rationale to include both results and discussion in the same section. Logic would say to include them as separate sections.

Lines 395-396: "a linear test.." – this belongs to the method section.

Line 397: what do you mean by "mechanical effect"?

The entire paragraph starting from line 395 and ending at422 needs to be moved to the methods section.

Line 402: in what way it is "inverted"? please specify this term within the scope of your methods.

Lines 406-407: what is the difference between passing vehicles ($P_v$) and environmental noise ($P_N$)?

Line 413: Equation (8). This is an interesting approach. However, I suggest you reference and cite Roth et al. (2016; https://doi.org/10.1002/2015JF003782), as they proposed a very similar model. Additionally, it is important to clearly state that $P_f$ represents the total contribution of all seismic sources. Moreover, you should explicitly mentions your key assumption that is currently implicit: the total seismic PSD, representing the seismic data at a given time and frequency, is assumed to be the linear sum of the PSDs of the contributing sources. This assumption underpins your presentation of Eq. (8), which, as it stands, is a reorganized form of this principle. This connection is not immediately evident from the text but is critical to understanding your approach (see Eq. (1) in Roth et al., 2016). Implicit as well in your Eq. (8) is that bedload transport did not occur during the monitoring period. While this is plausible, it should not come implicitly and as a surprise to the reader. Therefore, please add a sentence before introducing the equation,. Where you explicitly mention this. Lastly, please define the coefficient $a_f$.

Line 416: what do you mean by "highest turbulence signal to ratio"; what is "highest"? how is it defined?

Line 418: grammar is incorrect, please revise sentence.

Line 420-421: you mention the Green's function, but you assume that your audience is familiar with it, while it comes with no introduction. Introduce it, or delete.

**Figure 12:**

I propose to switch the axes, so that frequency is at the horizontal, while the coefficient is the vertical.

What is the meaning of the three panels? What is the difference between them and what do they represent in your scientific context?

Line 429: "After regressing… the first 10 minutes of..." I find the description of the monitoring period unclear at this stage of the manuscript. By this point, the reader should have a comprehensive understanding of the monitoring timeline, but that is currently not the case. I believe this issue stems from how the experiments were introduced earlier in the text. Please revisit and address my comments in the Methods section to provide a more detailed and transparent explanation of the experimental setup and monitoring period.

Line 432: in what way do the flow velocity measurements integrate into this part of your analysis? I do not seem to be able to follow your reasoning.

**Figure 13:**

Panel b: The 3-dimensional representation of the results is unclear and may be difficult for readers to interpret effectively. I recommend replacing it with a 2D plot, which would likely provide a clearer and more straightforward visualization of the data.

Line 479: "We cannot establish a correlation…" On what analysis is this statement based? Please refer explicitly to the relevant analysis or figure. If you are referring to Figure 11, I would argue that the measured discharge somewhat resembles the North component of the seismic power. However, this observation lacks a statistical criterion that would allow for an objective evaluation and support such a judgment.

Lines 483-485: "Real-time monitoring of the turbulence process…" This statement feels somewhat overstated in relation to your findings. While you have successfully established a quantitative correlation between a hydrodynamic parameter and seismic data, I believe it is not fully justified to claim that this enables "real-time monitoring" of the turbulence process. Based on the evidence presented, it seems that your work has not yet revealed specific features of the turbulence phenomena itself. I suggest revising this claim to better reflect the current scope and implications of your results.

Lines 487-501: This paragraph currently serves as a summary of the analysis and findings. However, I recommend revising it to go beyond summarizing and focus on highlighting the broader implications of your research. Specifically, you could emphasize the new avenues this

work opens for further exploration, as well as its potential relevance and applications beyond the context of your specific local findings. This would help underline the significance of your work and its contribution to the field.

**References**

Gimbert, F., Tsai, V.C., & Lamb, M. P. (2014). A physical model for seismic noise generation by turbulent flow in rivers. *Journal of Geophysical Research: Earth Surface*, 119, 1–30, doi:10.1002/2014JF003201.

Lagarde, S., Dietze, M., Gimbert, F., Laronne, J. B., Turowski, J. M., & Halfi, E. (2021). Grain-size distribution and propagation effects on seismic signals generated by bedload transport. *Water Resources Research*, 57, doi:10.1029/2020WR028700.

Luong, L., Cadol., D., Bilek, S., McLaughlin, M., Laronne, J.B. & Turowski, J.M. (2024). Seismic modeling of bedload transport in sandy gravel-bed alluvial channels. *Geophysical Research Letters*, 129, e2024JF007761. https://doi.org/10.1029/2024JF007761

Roth, D. L., Brodsky, E. E., Finnegan, N. J., Rickenmann, D., Turowski, J. M., & Badoux, A. (2016). Bed load sediment transport inferred from seismic signals near a river. *Journal of Geophysical Research: Earth Surface*, *121*(4), 725-747.

Schmandt, B., Aster, R. C., Scherler, D., Tsai, V. C., & Karlstrom, K. (2013). Multiple fluvial processes detected by riverside seismic and infrasound monitoring of a controlled flood in the Grand Canyon. *Geophysical Research Letters*, 40, 4858–4863, doi:10.1002/grl.50953.

Tsai, V.C., Minchew, B., Lamb, M. P., & Ampuero, J.P. (2012). A physical model for seismic noise generation from sediment transport in rivers. *Geophysical Research Letters*, 39, n/a-n/a, doi:10.1029/2011GL050255.

---

## Referee Comment (RC2)

**Review for egusphere-2024-4111, title: A physical model for mean river discharge calculation: from riverside seismic monitoring experiments in a low flow river, China**

Manuscript egusphere-2024-4111 entitled "A physical model mean river discharge calculation: from riverside seismic monitoring experiments in a low flow river, China" by Xiaoyue Zhou and co-authors investigates the relationship between seismic signal, streamflow properties and human-generated seismic noise along a reach of the Jiuqu river, China. The manuscript notably proposes a model to predict discharge based on the seismic energy recorded in the lower frequency domain (e.g. < 10 Hz).

I am not recommending to pursue the revision process of this manuscript in its current form, because I think it is too far from the standards of the journal Earth Surface Dynamics. A publication of the manuscript would require in my view a clearer definition of the study aim, a clearer identification of the knowledge gap to fill with regards to the literature, and clearer description of the methodology. In addition, the presented results are either not new (e.g. seismic energy related to turbulence in the lower frequency domain) or not convincing enough (e.g. the performance and necessity of the proposed empirical model). I am reporting general comments, specific comments and technical corrections in the section below to help the authors to follow my review and revise the manuscript.

**General comments**

The general structure of the manuscript is not satisfying. There is no clear cut between methodological considerations, manuscript results and result discussion, which are mixed up through sections 3 to 5.

The introduction does not justify well enough why the microseismic sensing of the streamflow is valuable as compared to more traditional hydrological stations or remote sensing methods. Those different approaches and their respective strengths and weaknesses should be more clearly identified in the introduction, with appropriate referencing, to better capture the rationale of the manuscript, which remains generally unclear. In this context, I would focus on streamflow seismic sensing – the main scope of the manuscript – and leave apart considerations about bedload transport seismic sensing.

The description of the experiments in the second section does not provide with enough details, and some more information about the set-up of the different experiments would be valuable to better follow the procedure, and for a matter of reproducibility. In addition, the distinction between different cross-section, experiments and tests is not always clear to me and makes the manuscript difficult to read. More importantly, the purpose of the different experiments should be made much clearer at an earlier stage, already at the end of the introduction, so that the reader exactly knows what will be done and why to tackle the research question.

Some of the presented results – e.g. seismic frequency domain of the water turbulence – are not new (e.g. Schmandt et al., 2013). In addition, I am not convinced that the precision of the flow velocity measurements really allow to use this dataset to test the performance of the seismic discharge model. In particular because there are so little variations in discharge during the experiment, as well as possible contamination of the seismic signal by people walking in the river to perform the flow velocity measurements (unless I miss something in the Method section). The proposed model to predict discharge based on

seismic sensing is empirical in my view, since it requires a calibration based on local measurements, and is therefore not physical as stated in the manuscript title. Instead, there is already a seismic physical model of turbulence (Gimbert et al., 2014) that has been published and tested (e.g. Dietze et al., 2019), and I am therefore wondering what the proposed model brings in comparison.

I find the bimodal distribution of seismic power (Figure 10a) in the frequency bands 2-15 Hz and 35-50 Hz for the steeper and potentially more turbulent section to be an interesting result. Effectively, this shows that there may be more overlap between turbulence and bedload transport frequency bands for more turbulent river, which may be an issue to invert bedload transport rates in this type of settings. I would suggest the authors to reorient the scope of their manuscript in this direction.

In general, the grounding of the manuscript into an appropriate literature context is insufficient, so is the mobilization of the literature to illustrate and support the argument, and to put the results into perspective in the Discussion. In many instances, the used references are irrelevant and/or erroneous with regards to the treated topic (e.g. reference to Aderhold et al., 2015 on l. 101-104, or to Bagnold et al., 1966 on l. 325). There are some articles in the bibliography that are not cited in the manuscript.

**Specific comments**

**1. Introduction**

**l. 33-40:** I do not find the first paragraph to be very efficient to justify the need for microseismic studies of water discharge.

**l.36:** Please add a few references to illustrate your argument about remote sensing technology.

**l.38-39:** Cook and Dietze (2022) and Larose et al. (2015) have reviewed environmental seismology. They may not be the best references to illustrate your point about time-consuming and resource-intensive approach of hydrological stations.

**l. 41:** Please define what you mean by "microseismic".

**l.42-43:** Turowski et al. (2011) have not used environmental seismology, but acoustic sensors.

**l. 46:** Rickenmann et al. (2012) have used acoustic monitoring of bedload transport, not microseismic.

**l. 57-69:** The focus should remain on water depth/discharge seismic sensing, and not much on bedload transport sensing, since it is not something done in the frame of this manuscript.

**l. 69:** Gaeuman (2014) is an off-topic reference.

**l. 71-78:** The paragraph goes in too many directions, without a clear structure, and lacks of appropriate scientific references.

**l.76-77:** I am surprised that the turbulence model of Gimbert et al. (2014) to invert streamflow information from low frequency band seismic energy is not mentioned, described and further mobilized here, and throughout the manuscript.

**l. 87:** Not convinced Viparelli et al. (2011) is an appropriate reference in this context.

**2. Experiments**

**l. 91-97:** Characteristics and statistics that are provided to describe the field site are not backed up with references.

**l. 97-98:** Can you provide some statistics on the discharge of the Jiuqu river (annual mean, variability, discharge during the experiments, etc.). Also, why is the discharge different between the different sections (e.g. tributary input?). If assuming mass conservation, the discharge is constant between the different sections, and it is its partitioning between width, depth and flow velocity that varies.

**l. 97-101:** At line 97, it is mentioned that "four monitoring experiments were conducted at four sections (…)". At lines 99-100, "we selected a curved section of the Jiuqu". I do not understand the link between the initial four sections, and the following one section.

**l. 101-102:** the description of the channel morphology could be clearer. What is meant by "drainage canal"?

**l. 101-104:** the river studied in Aderhold et al. (2015) is in New Mexico! How do you transfer their local grain-size distribution measurements to the Jiuqu River, China?

**l. 105-107:** Is this information about silt and sediment concentration needed with respect to streamflow seismic sensing? If it is, please justify its purpose in the text. If not, I would remove it.

**l. 107-108:** Would be good to have some information about mean annual streamflow (cf. comment l.97-98), to be able to scale it with respect to the streamflow during the experiment. Please add a reference to the concept of "low-flow river".

**l. 108:** The morphology of the different sections should be described more extensively than what is provided in Figure 2 or along l. 101-102. Why are those different channel morphologies selected with respect to the research question?

**l. 119:** Location of the seimic sensor is not clear.

**l.119-122:** I would say this information is not needed.

**l. 123-124:** Phrasing not clear enough with respect to which of the two types is sensitive to high-frequency (…). Please rephrase.

**l.125-132:** I am not convinced the content of this paragraph is of first relevance with regards to the focus of this manuscript.

**l. 142-143:** why isolating the sensors from the ground if seismic waves travelling in the ground are aimed to be measured?

**Figure 3:** I am struggling to follow between the four test sites in Figure 1, and the four seismic stations in Figure 3. Please clarify how many seismometers were used in each site, where there were located, what distance to the stream, what were the potential sources of external noise, etc. Perhaps a summary table would be helpful for this purpose.

**l. 154-155:** According to Figure 3, S3 is located about ~20 m from the riverbed, not 1 m.

**l. 158:** Is an 'experiment' the same thing than a 'test'. Please use the same word to refer to the same step, to easen the reading.

**l. 163-169:** This paragraph may belong to the 'Result' section.

**Figure 4:** I am surprised you measured no seismic signal related to turbulence at S4 and S5, while you remain very close to the river. It would be good to have the same y-scaling on the all four velocity plots. Isn't it also a problem of scaling that the seismic noise associated to turbulence does not appear on spectrograms where external noises are also present?

**l. 182:** Unclear what is meant by 'appropriate devices'. Please be more specific, with argument illustrated by references.

**Figure 5:** I am not sure the diagram is of first relevance with respect to the scope of the manuscript. It could go in a Supporting Information.

**Equations (1) to (7):** Please specify the units of every used variables.

**3. Seismic ambient noise**

**l. 220:** It is not clear if there should be the 'Results' section starting here, or whether we are still in the methodology description.

**l. 223:** Reference to back up your statement.

**l.224-225:** Not clear what "flow configuration of the river" means in this context. Please be more specific.

**l. 227:** Reference to back up your statement.

**l. 228-231:** In this paragraph, I would really specify what process can be distinguished (again, focusing on streamflow) in which frequency band, with referencing to the appropriate literature.

**l.241-246:** This paragraph may belong to the 'Results' section.

**l.246-248:** More justification about the signal filtration would be valuable (e.g. further arguments, a figure, some references).

**Figure 6:** as for Figure 4, it would be good to have the same y-scale on both ground velocity plots to ease the reading.

**l.262:** It is still not clear to me whether the different Tests are used to investigate the same thing at different sites, or whether different experiments are conducted at each site.

**l. 272-276:** Comparison of the 'Results' with the literature belongs to the Discussion section I think.

**l.278-280:** This is effectively no new result (e.g. Gimbert et al., 2014).

**l. 283:** Please be consistent and specific with the naming. What are locations 1, 2 and 3? According to l. 256-270, location 1 should instead be Test 1, S4; location 2 should be Test 2, S2, etc. I agree this does not read well, so I think there is a general rework to do to rename all the experiments and sensor location in a clearer way.

**l. 293-297:** Mainly repetitions from l. 221-227.

**l.298-299:** Those considerations should come earlier in the manuscript. The reference to Boano et al. (2011) is irrelevant (i.e. the paper is about hyporheic exchange, not sediment transport).

**Figure 9:** This is all known, not needed.

**l. 312-315:** This is an interesting result, that you get a bi-modal distribution of seismic energy for the river section that is more turbulent.

**l.317-328:** Again, I would say those considerations belong to the Discussion.

**l.320-322:** But you were in the field during the experiment. Was there any bedload transport going on? Perhaps your result shows that more turbulent section actually produce seismic energy in a higher frequency band, which partly overlaps with the frequency band we typically attribute to bedload transport.

**l. 325:** Why do you mention human activities here? Reference to Bagnold (1966) is inappropriate in this context. Not convinced the reference to Turowski and Bloem (2016) is neither relevant in this context.

**l.331-339:** Please specify in the Figure 10 caption for which Test and sensor location the different diagrams relate to.

**4. Seismic interpretation and river discharge calculation**

**l.342-364:** This is pure methodological description, and it should appear much earlier in the manuscript. You have already presented many result figures containing PSDs.

**l.370-374:** Assuming that the width and depth are constant then? But if there are changes in flow velocity, there may be changes in width and depth too, right?

**l.378-381:** I do not find the matching that clear. You may want to use a metric like a correlation to support your observation more quantitatively. The variations in discharge are very small, probably much smaller than the precision you get with the hydraulic estimate of discharge, so I am not convinced those results are robust enough. In addition, I guess your were walking into the river to do the velocity measurements, and this may have also been recorded in the seismic signal.

**Figure 11:** Why is it needed to present the results for every 3 components of the seismic sensor?

**5. Results and Discussion**

**l.396-397:** Please specify what you mean by "without considering the mechanical effects generated by the river process".

**l. 397-403:** Again, not clear from which experiment, test, location, section, etc. we are talking about.

**l. 403-404:** Not clear to me.

**l.420-421:** How were the parameters of the Green's function estimated?

**Figure 12:** Same question than for Figure 11: why is it needed to present the results for all 3 component of the seismic sensor? It is not clear to me what is presented in this Figure.

**l.434:** Why those frequencies in particular?

**l.437-439:** I do not see any comparison between the seismic prediction of discharge and the flow calculation using the flow velocity measurement in Figure 12. Perhaps an issue related to the labelling of the Figure? A plot showing measurements vs predictions would be a more efficient way to assess the performance of the model (as effectively done in Figure 13).

**l.442-443:** How the reported error magnitude compares to the variability in discharge during the experiments? I think the precision in the measurements using flow velocity does not really allow to test robustly the seismic model.

**l. 466-472:** In general, it looks like you are proposing an empirical model (and not a physical one as argued in the title) that requires calibration to predict discharge, while a physical model of turbulence has already been developed (Gimbert et al., 2014) and tested in multiple instances (e.g. Dietze et al., 2019), so I am not convinced by the usefulness of the proposed model.

**6. Conclusion**

**l. 474-476:** Again, not clear terminology to describe how the seismometers (number and location) are deployed.

**l.477-479:** It is expected that there are no correlation between the flow and seismic power at higher amplitudes.

**l. 481:** In what Result Figure do you observe a linear relationship?

**l.481-483:** This is not really a result from this study.

**l. 487-503:** The organization of the argument in this paragraph is very fuzzy.

**7. References**

**l. 516:** Burtin et al. (2008) not cited in the manuscript, while it definitely should.

**l. 536:** Foulds et al. (2014) not cited in the manuscript.

**Technical corrections**

**l. 33:** "usually include encompass (…)". Two different verbs saying the same, no?

**l.57-58:** "the vibrations of river sediment". Wording not ideal.

**l. 91:** "the river studied in this study". Poor wording.

**l. 176:** "precision" instead of 'accuracy' I think.

**l. 510-586:** in multiple instances (e.g. Schmandt et al., 2013), I saw that there are missing spaces between words in the references. Please check and correct throughout.

**l. 563:** Reference not sorted in alphabetic order.

**l.275:** Tsai et al. (2012), and not Tasi.

**l. 568:** Tsai et al. (2012) cited twice in the bibliography.

**References**

Aderhold, K., Anderson, K.E., Reusch, A.M., Pfeifer, M.C., Aster, R.C., Parker, T., 2015. Data Quality of Collocated Portable Broadband Seismometers Using Direct Burial and Vault Emplacement. Bulletin of the Seismological Society of America 105, 2420–2432. https://doi.org/10.1785/0120140352

Bagnold, R.A., 1966. An approach to the sediment transport problem from general physics. US government printing office.

Burtin, A., Bollinger, L., Vergne, J., Cattin, R., Nábělek, J.L., 2008. Spectral analysis of seismic noise induced by rivers: A new tool to monitor spatiotemporal changes in stream hydrodynamics. Journal of Geophysical Research: Solid Earth 113. https://doi.org/10.1029/2007JB005034

Cook, K.L., Dietze, M., 2022. Seismic Advances in Process Geomorphology. Annual Review of Earth and Planetary Sciences 50, 183–204. https://doi.org/10.1146/annurev-earth-032320-085133

Dietze, M., Lagarde, S., Halfi, E., Laronne, J.B., Turowski, J.M., 2019. Joint Sensing of Bedload Flux and Water Depth by Seismic Data Inversion. Water Resources Research 55, 9892–9904. https://doi.org/10.1029/2019WR026072

Foulds, S.A., Griffiths, H.M., Macklin, M.G., Brewer, P.A., 2014. Geomorphological records of extreme floods and their relationship to decadal-scale climate change. Geomorphology 216, 193–207. https://doi.org/10.1016/j.geomorph.2014.04.003

Gaeuman, D., 2014. High-Flow Gravel Injection for Constructing Designed in-Channel Features. River Research and Applications 30, 685–706. https://doi.org/10.1002/rra.2662

Gimbert, F., Tsai, V.C., Lamb, M.P., 2014. A physical model for seismic noise generation by turbulent flow in rivers. Journal of Geophysical Research: Earth Surface 119, 2209–2238. https://doi.org/10.1002/2014JF003201

Larose, E., Carrière, S., Voisin, C., Bottelin, P., Baillet, L., Guéguen, P., Walter, F., Jongmans, D., Guillier, B., Garambois, S., Gimbert, F., Massey, C., 2015. Environmental seismology: What can we learn on earth surface processes with ambient noise? Journal of Applied Geophysics 116, 62–74. https://doi.org/10.1016/j.jappgeo.2015.02.001

Rickenmann, D., Turowski, J.M., Fritschi, B., Klaiber, A., Ludwig, A., 2012. Bedload transport measurements at the Erlenbach stream with geophones and automated basket samplers. Earth Surface Processes and Landforms 37, 1000–1011.

Schmandt, B., Aster, R.C., Scherler, D., Tsai, V.C., Karlstrom, K., 2013. Multiple fluvial processes detected by riverside seismic and infrasound monitoring of a controlled flood in the Grand Canyon. Geophysical Research Letters 40, 4858–4863. https://doi.org/10.1002/grl.50953

Tsai, V.C., Minchew, B., Lamb, M.P., Ampuero, J.-P., 2012. A physical model for seismic noise generation from sediment transport in rivers. Geophysical Research Letters 39. https://doi.org/10.1029/2011GL050255

Turowski, J.M., Badoux, A., Rickenmann, D., 2011. Start and end of bedload transport in gravel-bed streams. Geophysical Research Letters 38. https://doi.org/10.1029/2010GL046558

Turowski, J.M., Bloem, J.-P., 2016. The influence of sediment thickness on energy delivery to the bed by bedload impacts. Geodinamica Acta 28, 199–208. https://doi.org/10.1080/09853111.2015.1047195

Viparelli, E., Gaeuman, D., Wilcock, P., Parker, G., 2011. A model to predict the evolution of a gravel bed river under an imposed cyclic hydrograph and its application to the Trinity River. Water Resources Research 47. https://doi.org/10.1029/2010WR009164

---

## Author Comment (AC2)

Dear Reviewers:

Thank you for your kind revisions and comments, in the revised version of the manuscript, we took into account all the comments of the reviewers and made the manuscript corrections as they suggested. In the revision submission, we provided two documents, one marked-up manuscript version (Xiaoyue_Zhou_REV. doc) showing the changes made, and the other final version of the manuscript (Xiaoyue_Zhou_DEF. doc). Thanks to the reviewers' kind suggestions, the structure of the paper was restructured in the revised manuscript, the discussion of seismic methods was reorganized in the introduction to ensure paragraph coherence, a detailed elaboration of the research objectives was added at the end of the introduction, the graphical questions raised were revised, the misuse of words and grammatical errors were carefully corrected, and the title of the manuscript was changed to "A empirical model for mean river discharge calculation: from riverside seismic monitoring experiments in a low-flow river, China" to better reflect the main content of the paper. In the REV file, corrections addressing the comments from Reviewer, are reported in red. All your suggestions are replied to as follows on a point-by-point basis.

**Comments (checklist) by the reviewers:**

**1.The manuscript, in its current form, lacks a clear motivation. Throughout the text, the authors' objectives are not sufficiently articulated. What is the main purpose of the project? Why were the experiments conducted? What was the rationale behind the choice of station locations? Additionally, why was the monitoring period limited to just one day? Please also refer to the last paragraph, after point (5).**

Response: Thank you for your questions. The main purpose of this study is to explore the application of seismic methods in river monitoring, and whether it is possible to monitor the discharge of rivers through seismic methods.

This experiment was conducted in a low-flow, remote mountainous river, which is far from busy traffic routes and residential areas, and has minimal human activity. Despite its remote location, it is easily accessible. Conducting the experiment in such a location helps to reduce the noise interference caused by human activities. The implementation of this project is primarily aimed at exploring the application of seismic monitoring technology in river flow monitoring. It involves obtaining seismic monitoring signals corresponding to different river discharge, analyzing their interrelationships, and constructing an empirical model for inferring river flow from seismic monitoring data.

Under the research objectives of this study, we conducted multiple different flow cross-section experiments within a single day, totaling four river seismic monitoring experiments. The experimental content fully meets the needs of this study. In addition, with the support of this research project, we also carried out long-term seismic monitoring experiments in another similar river, and more studies will be presented in the future.

**2.The structure of the paper is not well organized, which distracts the reading process. This may be partly due to an insufficient definition of the paper's scope (see my above point 1). The Introduction section lacks clear organization, with discussions of seismic methods interspersed throughout and paragraphs lacking coherence. Additionally, the focus on the actual flow**

**phenomena is insufficient and should be given more emphasis (see my comments on the first paragraph of the introduction). Moreover, many methodological details are only introduced in the Results and Discussion sections. I recommend a thorough reorganization of the manuscript, beginning with an introduction to flow phenomena and possibly a brief discussion on sediment transport. The authors can then identify research gaps and explain how environmental seismology can address these gaps. The research gaps must be then connected to the authors' current work.**

Response: Thank you for your suggestions. The structure of the paper was reorganized in the manuscript, with the introduction strengthening the coherence between paragraphs and focusing on the actual flow of the river, and the specific introductory revisions are shown in detail in the introduction section of the manuscript. The first paragraph of the introduction introduces the importance of river monitoring and the key phenomena of river flow, the second paragraph introduces the advantages of using seismic methods to monitor river processes, briefly mentions the key research in the field of river seismology, the third paragraph cites the specific research of scholars using seismic methods to monitor river processes to provide theoretical support for subsequent discussions, and the fourth paragraph introduces the purpose of this study and the contribution of current research to the field of river seismology. More details have been presented in the section "Introduction" to the manuscript.

**3.The figures are not evenly distributed throughout the manuscript and require revision. For example, the first three figures could be combined into a single, more cohesive figure. They should also better support the main text by helping the reader understand the rationale behind the seismic experiments. Additionally, schematic representations should be replaced with aerial or field photos (see my specific comments below).**

Response: Thank you for your suggestions. The figures mentioned by the reviewer in the manuscript that needs to be modified have been modified, the first three figures combined into a single, more cohesive figure. The schematic representations have been replaced with a field photos and using an aerial or satellite image to display the experimental sites along the river, as shown in Figure 1 in the manuscript. This restructuring improve the figure's effectiveness and enhance the manuscript's clarity.

[Figure]

Figure 1. The geophones at the four experimental sites. The red triangles represent the three base stations in test experiment 1, the green triangles represent the four stations in experiment 2, the blue triangles represent the four stations in experiment 3, and the yellow triangles represent the four stations in experiment 4.

**4.The empirical model used by the authors (Equation 8) closely resembles the one proposed by Roth et al. (2016), and this should be acknowledged. Additionally, the main title ("A physical model for...") does not align with the empirical nature of Equation 8. The authors should either provide a justification for why this model can be considered physical, or revise the title to more accurately reflect the empirical nature of the model and the main message of the paper.**

Response: Thank you for your suggestions. The model is indeed an empirical model, different from the physical model. Based on the research findings of Roth et al. (2016) and the hydrological and seismic data we obtained from the four experiments, modifications have been made to the model proposed by Roth et al. (2016). The new empirical model more directly and succinctly inverts the seismic data for calculation and is suitable for monitoring low-flow rivers. We have changed the title of the paper to "An empirical model for mean river discharge calculation: from riverside seismic monitoring experiments in a low-flow river, China," which better aligns with the research content of the paper and reflects the applicability of the empirical model. Thank you very much.

**5. The justification for the hydrological perspective needs to be strengthened. The authors introduce the field settings and describe the low-discharge regime during the survey period. They then calibrate their empirical model (Eq. 8) and apply it for discharge predictions. However, they should explicitly present the range of discharge values measured (or estimated using their approach) on the survey day. Specifically, what was the observed discharge range during the seismic and hydrological survey? If the fluctuations remain within a narrow range (e.g., within a factor of 1, meaning no substantial deviation from the average), then applying the model within the same range, even to unseen data, may have limited value. To address this, the authors should first present their discharge time series and then assess whether their model is applicable over that range. This clarification is crucial for strengthening the manuscript's narrative and overall scientific rigor.**

Response: Thank you for your suggestions. A description of the average annual rainfall at the study site and the range of river flow measured during the experiment were added to the overview of the experimental site to enhance the hydrological rationality. When the experiment was conducted on the same day, the measured river flow range was between 0.248 m³/s and 1.05 m³/s. The empirical model used in this study obtained the flow range value within the actual measured range value, and the average absolute error was between 0.03 and 0.2. Although there are still errors with the actual measured results, a more refined inversion model can be built in the future. More detailed have been presented in Line 126-128 in Section 2.1 "Experiment sites" in the Xiaoyue_Zhou_ REV file. Thank you very much.

**Specific comments**
**Introduction**
**I suggest restructuring the introduction, e.g., as follows:**

**1.First paragraph: Introduce the key phenomena of river flow and bedload transport, emphasizing their importance and the need for improved understanding.2. Second paragraph: Introduce environmental seismology as a remote sensing technique that mitigates risks associated with direct measurements of these processes. Briefly mention the key studies published in the realm of fluvial seismology, categorizing them by their focus (e.g., studies on flow, studies on bedload transport).3. Third paragraph: Use specific examples (e.g., Schmandt et al., 2013; Burtin et al., 2011; Roth et al., 2016) to provide a foundation for later discussions in the manuscript.4. Fourth paragraph: Identify the key research gaps that remain in the field and explain how the current study contributes to addressing these gaps. Emphasize how the manuscript provides a scientific evaluation of the proposed approach and its implications for advancing the understanding of river dynamics. This restructuring will improve the logical flow and strengthen the manuscript's narrative.**

Response: Thank you for your suggestions. The first paragraph of the introduction introduces the importance of river monitoring and the key phenomena of river flow, the second paragraph introduces the advantages of using seismic methods to monitor river processes, briefly mentions the key research in the field of river seismology, the third paragraph cites the specific research of scholars using seismic methods to monitor river processes to provide theoretical support for subsequent discussions, and the fourth paragraph introduces the purpose of this study and the contribution of current research to the field of river seismology. More detailed have been presented in Line 34-117 in Section 1 "Introduction" in the Xiaoyue_Zhou_ REV file. Thank you very much.

**Line 33-35: I suggest to include an opening sentence which can then be used to argue for the importance of river monitoring.**

Response: Done. An opening sentence which can then be used to argue for the importance of river monitoring has been added. 'Flood disaster is a common natural disaster, which occurs frequently in summer and autumn, and brings great harm to people's production and life. Therefore, it is of great practical significance to monitor rivers, strengthen flood control early warning and realize flood control and disaster reduction.'

**Line 34-35: there is also an empirical approach, where equations are used (e.g., for bedload transport and flow velocity).**

Response: Done. 'there is also an empirical approach, where equations are used'.

**Line 38-39: For the purpose of your argument, I would change the references of Roth, Cook and Larose, since these are directly Environmental Seismology studies, while your opening paragraph discusses river monitoring.**

Response: Thanks. The cited references have been changed." Both methods require the establishment of hydrological stations and the installation of measuring instruments, which makes the monitoring process complex, time-consuming, and resource-intensive (Hsu et al., 2011; Picozzi et al.,2023)." References have been adjusted.

**Line 40-42: The introduction of the seismic methods comes without a scope and I think you should first establish the reasoning for the use of seismic signals to infer river activity.**

Response: Thank you for your suggestions. As mentioned above, the impact of flood disaster on

human society requires monitoring of river process to avoid flood invasion. Compared with other river monitoring means, seismic method is more time-saving and labor saving. Therefore, this paper applies seismic method to river process monitoring.

**Line 43-36: Sentence is redundant, and I don't see how it contributes to your arguments in this paragraph. Either reformulate or emit.**
Response: Done. The sentence has been emitted.

**Line 46: "microseismic" – since this term repeats in your manuscript, and since it is used differently in most (if not all) the environmental seismology-related papers, I propose to either change it to "seismic signals" or to define what is "microseismic signal" (as opposed to a "seismic signal").**
Response: Done. I have changed "microseismic signal" to "seismic signals".

**Line 49: "water surface".**
Response: Thanks. Changed "water and the atmosphere" to "water surface".

**Line 52: what do you refer to by using "This"? be specific. I also find that your reasoning here lacks a strong argument, that is, "This demonstrated that … can not only aid in studying….", but the above sentence is not convincing enough. I propose to revise the sentences between lines 46 and 52 to better present the arguments (using the studies you mentioned – Schmandt and Diaz) leading to the motivation of using seismic instruments to understand river flow.**
Response: Thank you for this suggestion. The sentence you mentioned has been revised. The research of the above two scholars demonstrates that seismic monitoring can not only aid in studying the hydrological characteristics of rivers but also has significant potential in assessing hydrological hazards.

**Line 71-78: The entire paragraph reads detached from the rest. You begin with "warning systems…", then you go on discussing the general method of analysis ("by analyzing the time-frequency…"). but it feels a more general description of the method, so it needs to come earlier, where you introduce environmental seismology.**
Response: Done. The paragraph order has been adjusted.

**Line 80: "This study focuses on the monitoring…" – I am lacking a motivation description to your study. This should come in the above paragraph. In what way does your experiment adds on previous experiments that conducted field experiments with flow and seismic measurements? Is it related to the flow regime in your study sites? Is it related to the number of your monitoring sites? Explain.**
Response: Thank you for your suggestion, the statement of the research objectives may not be clear enough. The last paragraph introduces the research objectives. Seismic methods are used to monitor river processes, and river flow inversion models are constructed based on seismic data and flow data, so as to provide references for river flood monitoring and early warning in this region and downstream river flow changes.

**2 Experiments**

**Line 89: I propose to change the title from "Experiments" to "Methods".**

Response: Done.

**Line 91: "We studied the Jiuqu River, a tributary of…"Lines 92-93: Instead of refereeing to "Meishui town", refer to either the river or the catchment itself.**

Response: Done. It has been revised." The river studied in this study, the Jiuqu River, located in the territory of Meishui Township in Shangyou County, China (Figure 1)."

**Line 93: what is a relative height? Do you mean Relief?**

Response: Thanks, I have changed "relative height" to "Relief".

**Line 94-95: Is the description of lithology important to your scientific work presented here? If it does not serve an important information, I propose to remove it.**

Response: Done." description of lithology" has been removed.

**Line 97: either "four field monitoring experiments", or "four field experiments"**

Response: Thanks." In this study, four field monitoring experiments were conducted at four sections of the Jiuqu River with different discharge."

**Line 98: what are current meters? How do these measure flow velocity of the flow when situated o the banks?**

Response: Thanks, "current meters" is "Flowmeters".

**Line 99: "for this experiment", what do you mean by "this?"**

Response: Thanks, Mistranslated, "this" refers to the entire study. It has been revised in the manuscript.

**Line 102: I am unclear what is a drainage canal.**

Response: Mistranslated, I have changed "drainage canal" to "braided river".

**Line 104: when describing grain size, please refer to standard jargon, for example, using its axis (axis a, b and c). Line 105: I think you would benefit from reporting the average length of axis b itself, rather than the volume.**

Response: Done. I revised it in the manuscript." The axes a, b and c of the maximum gravel are 50,36 and 20cm respectively, the average length of axis b is 10cm."

**Line 106: Was your sediment concentration meant for the purpose of evaluating sediments travelling in suspension? Clarify? From what elevation above the bed were these measurements conducted?**

Response: Thank you for your questions. I removed the sediment concentration.

**Line 108: "which classifies it as a low-flow river" – according to what standards? Please give reference, or revise\emit.**

Response: Done. I emitted it.

**Line 119: "seismic ambient noise was collected" – the noise was not collected. Measurements were taken, or seismic noise was monitored. Please revise. Lines 119:121: "Seismic instruments offer a variety of…" – delete this sentence.**
Response: Done. It has been revised." Seismic ambient noise was monitored from both the river sections and the nearby road areas."

**Line 134: not seismic stations, but seismic instruments.**
Response: Done." seismic stations" was changed "seismic instruments".

**Line 150: "Match 17" – the monitoring period was for one day? What was your purpose in conducting an experiment for one day? How long within that day did your experiment last?**
Response: The study lasted for one day because this experiment was a preliminary exploration of the application of seismic methods to monitor the river process. Four river sections were selected in this river, and the monitoring time of each section was 20 minutes. Long-term monitoring experiments will be conducted in the future for more detailed analysis.

**Lines 150-151: move this to the beginning of the Method section to emphasize your objectives.**
Response: Thank you for this suggestion. I moved this sentence to the beginning of the Method section.

**Line 158: what is the second experiment? I find it a little bit challenging to find my way within the introduction of your method section structure. For example, when you write "For the second experiment…" would imply for a reader that there are a few experiments. In such a case, you need to introduce them properly, before you delve into explaining them in detail.**
Response: Thank you for this suggestion. This study selected a total of four river segments, each corresponding to a small experiment, these four river segments of the small experiment constituted an experiment of this study.

**Line 163: Why did you use four stations? In Figure 3, it seems that the stations are very close to each other. What were your aims in deploying for, and not only one stations? This needs to be justified.**
Response: Thanks. Four stations are used to better monitor the seismic signals generated by the river and the noise generated by the traffic, one station is used to monitor the noise signal of the vehicle, and the remaining three are used to monitor the seismic signal generated by the river.

**Lines 163 – 169: I find this paragraph more suitable for the Results section.**
Response: Thanks. I revised it in manuscript.

**Line 180: "calculating river flow" – what is river flow? Please specify, is that discharge (e.g., in units of m3/s) or is it flow velocity (e.g., in units of m/s)? Line 186: again, I think you mean here "discharge" rather than "river flow".**
Response: Done. "river flow" is "discharge".

**Line 193: the title feels detached. Revise to make clear what you are doing.**

Response: Done. I changed "River flow velocity measurement and discharge calculation" to "Discharge calculation"

**Line 218: "tests" – do you mean experiments? Please be consistent with the terms you use.**

Response: Done. I changed "tests" to "experiments".

**3 Seismic ambient noise**

**Line 220: change title to "Results". It was not clear from the title that this is the results section, and I was reading a substantial part of this section believing it is still the methods (see comments below).**

Response: Thank you for this suggestion. The structure and logic were reorganized in the manuscript and this section was moved to the methods section.

**Lines 221-222: This comment applies specifically to the mentioned sentence but can be generalized to similar instances throughout the manuscript. It is crucial to maintain a clear distinction between the Introduction, where you present the state of the science, previous studies, and relevant fields, and the Methods, where you concisely describe the techniques used, ensuring they are theoretically reproducible. For example, in this case, rather than directly explaining the seismic ambient noise method, lines 221–231 primarily serve to justify the approach. However, such rationale belongs in the Introduction rather than the Methods section. Ensuring this distinction throughout the manuscript will improve clarity and logical structure.**

Response: Thank you for this suggestion, throughout the manuscript, the structure has been rearranged according to the suggestions, improving the clarity and logical structure of the manuscript, and the specific corrections are shown in detail throughout the manuscript.

**Line 239: please specify the distance of the road from the experimental sites.**

Response: Done. This part has already been added to the manuscript. "The river section studied in this experiment is located next to a road. The distance between the road and the experiment site is 200m."

**Figures 1, 2 and 3: I believe the manuscript would benefit from a more integrated and visually informative representation of the experimental sites. Instead of separate schematic cross-sections and contour- like maps, I suggest merging Figures 1, 2, and 3 into a single figure with multiple panels. This would provide a more comprehensive overview of the study area by: (1) including a small inset map to show the study location within a broader China map for geographic context, (2) using an aerial or satellite image to display the experimental sites along the river, (3) incorporating photos of the different study sites to highlight their distinct morphological characteristics, and (4) ensuring a clearer depiction of station locations within the field setting, as the current photo in Figure 3 does not make their placement evident. Additionally, the sensors in Figure 3 appear wired to cables - are these temporary? More details on the station setup, location, and monitoring period would improve clarity. Lastly, the red sensor between S2 and S4 lacks a label— - please clarify its designation. This restructuring**

**will improve the figure's effectiveness and enhance the manuscript's clarity.**

Response: Thank you for this suggestion. In the manuscript, Figure 1, Figure 2 and Figure 3 were removed, and satellite maps and field experiment photos were used to show the experimental site setting for this study.

[Figure]

Figure 1. The geophones at the four experimental sites. The red triangles represent the three base stations in test experiment 1, the green triangles represent the four stations in experiment 2, the blue triangles represent the four stations in experiment 3, and the yellow triangles represent the four stations in experiment 4.

**Figure 5: In the figure's caption, please explain all the variables you mention in the figure.**

Response: Done. The variables have been explained in the diagram.

**Figure 6: I believe this figure is redundant in the main text as it is secondary to your main analysis. Therefore, I propose to move it to the supplementary information.**

Response: Thank you for this suggestion. This figure has been removed from the body of the manuscript.

**Line 255: What is "still water flow"?**

Response: "still water flow" is "transient flow".

**Lines 256-257: please specify "experiment" and "test". Are there differences between these terms? Are these the same?**

Response: Thank you for your questions. These two represent the same meaning and have been revised in the manuscript.

**Lines 257-260: This sentence is redundant. Delete.**

Response: Thanks. It has been deleted.

**Line 260: "A clear broadband seismic response" – in what way it is clear? That is subjective.**

**I would try to reformulate the sentence to a more objective one.**

Response: Thank you for your suggestion. A more objective formulation of this phrase is given in the manuscript.

**Lines 272-280: This entire paragraph in an interpretation, not a result. Please move it to the discussion if needed.**

Response: Done. The order of the paragraphs in this sentence has been adjusted to the discussion section.

**Figure 8: "Acceleration power spectral" – why do you use acceleration rather than velocity? "PSD plots" – I think these plots would be better called "seismic spectra". Why do you choose to exhibit three curves and not all of the curves that you have as data sets?**

Response: Thanks. The title of this graph was changed to "seismic spectra", and the E-channel data with the highest signal-to-noise ratio among the three stations monitoring the water flow signal was selected, which was displayed more clearly.

**Lines 293-299: Again, this paragraph belongs to the introduction. Remember that you are now introducing your results, so at this stage the reader should already be informative of the fact that "geophones can detect elastic waves generated by processes…". In fact, you already wrote this previously, so it is a repetition. In this manner, the allocation of Figure (9) is not well suited**

Response: Thank you to the reviewers for their careful review. After careful consideration, this passage has been removed from the manuscript and Figure 9 has been deleted.

**Lines 300-302: You are mixing pure results (e.g., the measurement showed that…) with discussion, which is meant to bring forth your interpretation of the results (e.g., "the deployed microseismic stations can receive these signals and record them as").**

Response: Thanks. This question has been revised in the manuscript and the language has been reorganized to present the results part.

**Lines 310-312: You introduce the fourth experiment, but this should already have been done previously in the methods section.**

Response: Thank you for this suggestion. The presentation of this fourth experiment has been adapted to the methods section.

**4 Seismic interpretation and river discharge calculation**
**Line 340: please change this title to "Discussion.**

Response: Done. I changed this title to "Discussion.

**Lines 356-364: Same comment as above. This paragraph should be deleted. The information is interesting, but it is not related with your specific work. The focus should be on interpreting the results, not the methods, which are probably massively discussed elsewhere in the literature.**

Response: Thanks. This passage has been removed from the manuscript.

**Lines 367-368: The sentence is superfluous.**
Response: Thanks. It has been deleted.

**Lines 368-370: It is too early to state that. You should begin with you aim, proceed with what you did as a test (as you begin in line 370, "for verification"), then conclude with something in the form of "as a result, we propose that the frequency range within 2 and 10 Hz records seismic signals generated form turbulence…". Alternatively, you could write something along the lines of: "We hypothesize that the low- frequency… in the experimental … is related to flow turbulence within the stream". To validate our hypothesis, we plot…".**
Response: Thank you for this suggestion. This sentence has been rephrased, specifically as follows: "We hypothesize that the low-frequency band of 2~10 Hz in the experimental is related to flow turbulence within the stream to validate our hypothesis, for verification, we selected data from the third experiment."

**Figure 11: This figure should be within the results, not discussion. The discussion n can pick this up by discussing in what ways does this plot reinforces (or not) your hypothesis that turbulence dominates the < 10 Hz frequency band.**
Response: Thank you very much for this suggestion. Moving Figure 11 to the results section, the river flow in Figure 11 is obtained by the seismic frequency inversion of 2-7 Hz, and it can be seen from the figure that the inverted flow rate is in good agreement with the actual measured flow, indicating that the dominant frequency of the river seismic signal in this study is less than 10 Hz.

**5 Results and discussion**
**Line 394: it does not make much rationale to include both results and discussion in the same section. Logic would say to include them as separate sections.**
Response: Done. The results have been separated from the discussions, and the concrete presentation is reflected in the manuscript.

**Lines 395-396: "a linear test.." – this belongs to the method section.Line 397: what do you mean by "mechanical effect"? The entire paragraph starting from line 395 and ending at422 needs to be moved to the methods section.**
Response: Thank you for this suggestion. This section on linear models has been moved to the Methods section 2.3, which can be found in lines 253-280 of the manuscript

**Line 402: in what way it is "inverted"? please specify this term within the scope of your methods.**
Response: Thank you. Phrase usage error, I changed"inverted"to"inversed"。

**Lines 406-407: what is the difference between passing vehicles (Pv) and environmental noise (PN)?**
Response: "PN" refers to the noise signal in the environment when there is no vehicle noise, and "PN" has been annotated in the manuscript.

**Line 418: grammar is incorrect, please revise sentence.**

Response: Done. Modifications have been made. Specific modifications such as: "The flow regression coefficients for geophones with the highest turbulence signal-to-noise ratios, corresponding to each component (E, N, and Z) and each experiment site (Experiment 1, 2, and 4), calculated using the first 10 minutes of seismic data, are shown in Figure 12."

**Line 420-421: you mention the Green's function, but you assume that your audience is familiar with it, while it comes with no introduction. Introduce it, or delete.**
Response: Done. "Green's function" was emitted.

**Line 432: in what way do the flow velocity measurements integrate into this part of your analysis? I do not seem to be able to follow your reasoning. Figure 13: Panel b: The 3-dimensional representation of the results is unclear and may be difficult for readers to interpret effectively. I recommend replacing it with a 2D plot, which would likely provide a clearer and more straightforward visualization of the data.**
Response: Thank you for this suggestion. Figure has been modified as follows:

[Figure]

Figure 11. The plot of measured flow values against inverted flow predictions, (a). The scatter plot of mean absolute error between flow predictions and measured values, (b). In figure (b), red represents measured values and purple represents predicted values.

**Figure 12: I propose to switch the axes, so that frequency is at the horizontal, while the coefficient is the vertical. What is the meaning of the three panels? What is the difference between them and what do they represent in your scientific context?**
Response: Done.

[Figure]

Figure 10. The flow regression coefficients and 95% confidence intervals for the river processes. (a), (b), and (c) show the flow regression coefficients and 95% confidence intervals for the river processes on the E, N, and Z components of the ground motion at base stations SZ4, SZ2, and SZ4 at experiment 1, 2, and 4, respectively.

**Line 429: "After regressing… the first 10 minutes of…" I find the description of the monitoring period unclear at this stage of the manuscript. By this point, the reader should have a comprehensive understanding of the monitoring timeline, but that is currently not the case. I believe this issue stems from how the experiments were introduced earlier in the text. Please revisit and address my comments in the Methods section to provide a more detailed and**

**transparent explanation of the experimental setup and monitoring period.**

Response: Thank you for this suggestion. The experimental part of the method section has been described in more detail based on your opinion. "In this study, four field monitoring experiments were conducted at four sections of the Jiuqu River with different discharge. Flowmeters and seismic stations were installed on the riverbank to measure the flow velocity and seismic ambient noise in each segment. Four seismic monitoring devices were utilized for this study. The station deployment protocol was as follows: During each experiment, the S3 (Station 3) unit was consistently deployed near roadsides with frequent human activity. The remaining three sensors were distributed in different hydrological environments - one installed in the river channel, another along the riverbank, and a third positioned approximately 50 meters offshore for ambient environmental noise comparison monitoring. (Note: The offshore deployment was omitted during the fourth experiment due to site constraints.). Each experiment lasted for 20 minutes. Therefore, in all four experiments, the S3 (Station 3) was placed about 1 meter from the riverbank, near the road. This configuration aimed to record seismic signals generated by river activities while minimizing interference from human activities. The flow velocity was continuously measured at a sampling frequency of once per minute."

**Line 479: "We cannot establish a correlation…" On what analysis is this statement based? Please refer explicitly to the relevant analysis or figure. If you are referring to Figure 11, I would argue that the measured discharge somewhat resembles the North component of the seismic power. However, this observation lacks a statistical criterion that would allow for an objective evaluation and support such a judgment.**

Response: Thank you for this suggestion. This part has been revised in the manuscript, and the specific changes are shown in the manuscript. "We found that the signals generated by the river flow have a very wide frequency range (2~50Hz). The recorded seismic signals in the 2~10 Hz band have a strong connection with river flow, approximately exhibiting a linear relationship."

**Lines 483-485: "Real-time monitoring of the turbulence process…" This statement feels somewhat overstated in relation to your findings. While you have successfully established a quantitative correlation between a hydrodynamic parameter and seismic data, I believe it is not fully justified to claim that this enables "real-time monitoring" of the turbulence process. Based on the evidence presented, it seems that your work has not yet revealed specific features of the turbulence phenomena itself. I suggest revising this claim to better reflect the current scope and implications of your results.**

Response: Thank you for this suggestion. This sentence has been rephrased. "Even when the river flow is low, the correlation between seismic signals and turbulent processes can be discovered through analysis."

**Lines 487-501: This paragraph currently serves as a summary of the analysis and findings. However, I recommend revising it to go beyond summarizing and focus on highlighting the broader implications of your research. Specifically, you could emphasize the new avenues this work opens for further exploration, as well as its potential relevance and applications beyond the context of your specific local findings. This would help underline the significance of your work and its contribution to the field.**

Response: Thank you for this suggestion. The contribution of this study to this field is added to the

conclusion. e.g. "This discovery is conducive to further exploration of the use of seismic methods for monitoring river processes, establishing a more complete river flow inversion model, and better realizing the real-time monitoring of river processes through seismic methods."

**References**

Gimbert, F., Tsai, V.C., & Lamb, M. P. (2014). A physical model for seismic noise generation by turbulent flow in rivers. Journal of Geophysical Research: Earth Surface, 119, 1–30, doi:10.1002/2014JF003201.

Roth, D. L., Brodsky, E. E., Finnegan, N. J., Ricken man n, D., Turowski, J. M., & Ba doux, A. (2016). Bed load sediment transport inferred from seismic signals near a river. Journal of Geophysical Research: Earth Surface, 121(4), 725-747.

Schmandt, B., Aster, R. C., Scherler, D., Tsai, V. C., & Karlstrom, K. (2013). Multiple fluvial processes detected by riverside seismic and infrasound monitoring of a controlled flood in the Grand Canyon. Geophysical Research Letters,40, 4858–4863, doi:10.1002/grl.50953.

Tsai, V.C., Minchew, B., Lamb, M. P., & Ampuero, J.P. (2012). A physical model for seismic noise generation from sediment transport in rivers. Geophysical Research Letters, 39, n/a-n/a, doi:10.1029/2011GL050255.

Burtin, A., Bollinger, L., Vergne, J., Cattin, R., Nábělek, J.L., 2008. Spectral analysis of seismic noise induced by rivers: A new tool to monitor spatiotemporal changes in stream hydrodynamics. Journal of Geophysical Research: Solid Earth 113.https://doi.org/10.1029/2007JB005034

---

## Author Comment (AC3)

Dear Reviewers:

Thank you for your kind revisions and comments, in the revised version of the manuscript, we took into account all the comments of the reviewers and made the manuscript corrections as they suggested. In the revision submission, we provided two documents, one marked-up manuscript version (Xiaoyue_Zhou_REV. doc) showing the changes made, and the other final version of the manuscript (Xiaoyue_Zhou_DEF. doc). Thanks to the reviewers'kind suggestions. In the manuscript, the organizational structure of the paper is adjusted in detail, especially in the introduction part, which explains the advantages and disadvantages of seismological methods and traditional hydrological stations and remote sensing methods for monitoring river processes, and adds references for theoretical support, and describes the experimental settings in more detail, and draws a table of the placement of seismic instruments in the experimental process, which is composed of four experiments in four sections of a river, in order to explore the application of seismological methods in monitoring river processes. In this study, an empirical model was constructed based on the research results of previous scholars to invert the river flow, which opened up a good situation for us to explore the use of seismology to monitor the river process. In the REV file, corrections addressing the comments from Reviewer, are reported in red. All your suggestions are replied to as follows on a point-by-point basis.

**Comments (checklist) by the reviewers:**
**General comments**

**The general structure of the manuscript is not satisfying. There is no clear cut between methodological considerations, manuscript results and result discussion, which are mixed up through sections 3 to 5. In general, the grounding of the manuscript into an appropriate literature context is insufficient, so is the mobilization of the literature to illustrate and support the argument, and to put the results into perspective in the Discussion. In many instances, the used references are irrelevant and/or erroneous with regards to the treated topic (e.g. reference to Aderhold et al., 2015 on l. 101-104, or to Bagnold et al., 1966 on l. 325). There are some articles in the bibliography that are not cited in the manuscript.**

Response: Thank you for your kind revisions and comments. The overall structure of the manuscript has been reconstructed, and the introduction is reorganized, according to the importance of river monitoring and the key phenomena of river flow, followed by the advantages of using seismic methods to monitor river processes, the key research in the field of river seismology is briefly mentioned, and the specific research of scholars using seismic methods to monitor river processes is cited to provide theoretical support for the subsequent discussion, and finally the purpose of this study and the contribution of current research to the field of river seismology are introduced. This study consists of four experiments, four river sections are selected from the upstream to the lower reaches of the river, and each section is subjected to a 20-minute river process monitoring experiment, and an empirical model is constructed to invert the river flow according to the previous scholars' research, and whether it is feasible to use seismic methods to monitor the river process. In this paper, literature in the field of river seismology in recent years has been added to the manuscript, and the research results of scholars are used to provide theoretical support for this research. Corrections were made for incorrect citations of references in the manuscript. Thank you very much.

**Specific comments**

**1. Introduction**

**l. 33-40: I do not find the first paragraph to be very efficient to justify the need for seismic studies of water discharge.**

Response: Thank you for your suggestions. The first paragraph of the introduction begins with an opening statement to demonstrate the importance of monitoring river processes and to describe the advantages of using seismology to monitor river processes. More detailed have been presented in Line 34-36 in Section 1 "Introduction" in the Xiaoyue_Zhou_ REV file. Thank you very much.

**l.36: Please add a few references to illustrate your argument about remote sensing technology.**

Response: Done. It has been presented in Line 42 in Section 1 "Introduction" in the Xiaoyue_Zhou_ REV file. Thank you very much.

**l.38-39: Cook and Dietze (2022) and Larose et al. (2015) have reviewed environmental seismology. They may not be the best references to illustrate your point about time-consuming and resource-intensive approach of hydrological stations. l. 87: Not convinced Viparelli et al. (2011) is an appropriate reference in this context.l.42-43: Turowski et al. (2011) have not used environmental seismology, but acoustic sensors. l. 46: Ricken man n et al. (2012) have used acoustic monitoring of bedload transport, not microseismic. l. 69: Gae u man (2014) is an off-topic reference.**

Response: Thank you for your suggestions. References have been adjusted.

**l. 41: Please define what you mean by "microseismic".**

Response: Thank you for your suggestions. "microseismic" refers to seismic signal, which has been changed in the manuscript.

**l. 57-69: The focus should remain on water depth/discharge seismic sensing, and not much on bedload transport sensing, since it is not something done in the frame of this manuscript.**

Response: Thank you for your suggestions. This paragraph has been rephrased.

**l. 71-78: The paragraph goes in too many directions, without a clear structure, and lacks of appropriate scientific references.**

Response: Thank you for your suggestions. The language of this paragraph has been reorganized and references have been added.

**l.76-77: I am surprised that the turbulence model of Gimbert et al. (2014) to invert streamflow information from low frequency band seismic energy is not mentioned, described and further mobilized here, and throughout the manuscript.**

Response: Done. The relevant content of this document has been cited and displayed in the manuscript.

**2. Experiments**

**l. 91-97: Characteristics and statistics that are provided to describe the field site are not**

**backed up with references.**

Response: Done. Added references to site feature descriptions.

**l. 97-98: Can you provide some statistics on the discharge of the Jiuqu river (annual mean, variability, discharge during the experiments, etc.). Also, why is the discharge different between the different sections (e.g. tributary input?). If assuming mass conservation, the discharge is constant between the different sections, and it is its partitioning between width, depth and flow velocity that varies.**

Response: Thank you for this suggestion. Data on the study of river flow have been added to the manuscript, and the specific changes are presented in the manuscript. The discharge different between the different sections may be due to the tributaries. The average annual precipitation in the study area was plotted.

[Figure]

Figure 2. Monthly precipitation at the experiment site

**l. 97-101: At line 97, it is mentioned that "four monitoring experiments were conducted at four sections (…)". At lines 99-100, "we selected a curved section of the Jiuqu". I do not understand the link between the initial four sections, and the following one section.**

Response: Thanks. The specific description of the four river monitoring experiments on the Jiuqu River has been revised in the manuscript.

**l. 101-104: the river studied in Aderhold et al. (2015) is in New Mexico! How do you transfer their local grain-size distribution measurements to the Jiuqu River, China?**

Response: Thank you for your suggestions, the references are misplaced and have been revised in the manuscript of the text.

**l. 105-107: Is this information about silt and sediment concentration needed with respect to streamflow seismic sensing? If it is, please justify its purpose in the text. If not, I would remove**

**it.**
Response: Thanks. The sediment content has been removed.

**l. 119: Location of the seimic sensor is not clear.**
Response: Thanks. The location of the seismic sensor is described in detail below. "In this study, four field monitoring experiments were conducted at four sections of the Jiuqu River with different discharge. Flowmeters and seismic stations were installed on the riverbank to measure the flow velocity and seismic ambient noise in each segment. Four seismic monitoring devices were utilized for this study. The station deployment protocol was as follows: During each experiment, the S3 (Station 3) unit was consistently deployed near roadsides with frequent human activity. The remaining three sensors were distributed in different hydrological environments - one installed in the river channel, another along the riverbank, and a third positioned approximately 50 meters offshore for ambient environmental noise comparison monitoring. (Note: The offshore deployment was omitted during the fourth experiment due to site constraints.). Each experiment lasted for 20 minutes. Therefore, in all four experiments, the S3 (Station 3) was placed about 1 meter from the riverbank, near the road. This configuration aimed to record seismic signals generated by river activities while minimizing interference from human activities. The flow velocity was continuously measured at a sampling frequency of once per minute."

**l.119-122: I would say this information is not needed.**
Response: Done. This sentence has been deleted.

**l. 123-124: Phrasing not clear enough with respect to which of the two types is sensitive to high-frequency (…). Please rephrase.l.125-132: I am not convinced the content of this paragraph is of first relevance with regards to the focus of this manuscript.**
Response: Done. The description of the instrument in these two manuscripts has been modified, and the specific changes are shown in the manuscripts.

**l. 142-143: why isolating the sensors from the ground if seismic waves travelling in the ground are aimed to be measured?**
Response: Thank you for this suggestion. This is the leveling bracket for the seismic instrument, and the seismograph needs to be used after leveling.

**Figure 3: I am struggling to follow between the four test sites in Figure 1, and the four seismic stations in Figure 3. Please clarify how many seismometers were used in each site, where there were located, what distance to the stream, what were the potential sources of external noise, etc. Perhaps a summary table would be helpful for this purpose.**
Response: Thank you for your suggestion. A table summarizing the location distribution and number of individual instruments is as follows:

Table1. List of the seismic stations used in this study and of their features in terms of sample rate, distance from the river and distance from the road.

| Name | Experiment | Sample Rate (Hz) | Distance from the River (m) | Distance from the Road (m) |
|---|---|---|---|---|
| S3 | 1 | 200 | 50 | 1 |
| S4 | 1 | 200 | 0.1 | 50 |

| | | | | |
|---|---|---|---|---|
| S5 | 1 | 200 | 1.5 | 50 |
| S2 | 2 | 200 | 0.1 | 50 |
| S3 | 2 | 200 | 100 | 1 |
| S4 | 2 | 200 | 1.5 | 100 |
| S5 | 2 | 200 | 3 | 100 |
| S2 | 3 | 200 | 1.5 | 100 |
| S3 | 3 | 200 | 100 | 1 |
| S4 | 3 | 200 | 0.1 | 100 |
| S5 | 3 | 200 | 3 | 100 |
| S2 | 4 | 200 | 1.5 | 100 |
| S3 | 4 | 200 | 100 | 1 |
| S4 | 4 | 200 | 0.1 | 100 |
| S5 | 4 | 200 | 3 | 100 |

**l. 158: Is an 'experiment' the same thing than a 'test'. Please use the same word to refer to the same step, to easen the reading.**

Response: Thank you for your suggestions, the two have been reconciled in the manuscript.

**l. 163-169: This paragraph may belong to the 'Result' section.**

Response: Done. This section has been moved to the results section.

**l. 182: Unclear what is meant by 'appropriate devices'. Please be more specific, with argument illustrated by references.**

Response: Done. I changed 'appropriate devices' to' flowmeters'.

**3. Seismic ambient noise**

**l. 220: It is not clear if there should be the 'Results' section starting here, or whether we are still in the methodology description.**

Response: Thanks. Adjust the order of paragraphs in the manuscript.

**l. 223: Reference to back up your statement.**

Response: Done. References have been cited.

**l.224-225: Not clear what "flow configuration of the river" means in this context. Please be more specific.**

Response: Thank you for this suggestion. "flow configuration of the river" refers to the shape and manifold of the river.

**l. 228-231: In this paragraph, I would really specify what process can be distinguished (again, focusing on streamflow) in which frequency band, with referencing to the appropriate literature.**

Response: Done. Specific frequency bands have been added to the manuscript and relevant references have been cited.

**l.241-246: This paragraph may belong to the 'Results' section.**

Response: Done. Adjustments were made to the paragraph distribution in the manuscript.

**l.262: It is still not clear to me whether the different Tests are used to investigate the same thing at different sites, or whether different experiments are conducted at each site.**

Response: Thanks. Each experiment is described, and this study consists of four experiments in these four river sections.

**l. 272-276: Comparison of the 'Results' with the literature belongs to the Discussion section I think.**

Response: Done. This paragraph has been placed in the discussion section.

**l. 283: Please be consistent and specific with the naming. What are locations 1, 2 and 3? According to l. 256-270, location 1 should instead be Test 1, S4; location 2 should be Test 2, S2, etc. I agree this does not read well, so I think there is a general rework to do to rename all the experiments and sensor location in a clearer way.**

Response: Done. The interpretation of the numbers of the individual instruments is specified in the table 1.

**l. 293-297: Mainly repetitions from l. 221-227.**

Response: Thank you for your suggestions, this part has been removed from the manuscript.

**l.298-299: Those considerations should come earlier in the manuscript. The reference to Boano et al. (2011) is irrelevant (i.e. the paper is about hyporheic exchange, not sediment transport).**

Response: Thank you for your suggestion. This sentence was removed from the manuscript after careful consideration.

**l.317-328: Again, I would say those considerations belong to the Discussion.**

Response: Thank you for this suggestion. It has been moved to the discussion section, which is presented in the manuscript.

**l. 325: Why do you mention human activities here? Reference to Bagnold (1966) is inappropriate in this context. Not convinced the reference to Turowski and Bloem (2016) is neither relevant in this context.**

Response: Thank you for your questions. Human activity is mentioned because this high-frequency band noise can be caused by human activity. The references have been revised.

**l.331-339: Please specify in the Figure 10 caption for which Test and sensor location the different diagrams relate to.**

Response: Done. In the title of Figure 10, different plots have been added in relation to which test and sensor position is relevant.

**4. Seismic interpretation and river discharge calculation**

**l.342-364: This is pure methodological description, and it should appear much earlier in the manuscript. You have already presented many result figures containing PSDs.**

Response: Thank you for this suggestion. This passage has been moved to the methodology section in the manuscript.

**l.378-381: I do not find the matching that clear. You may want to use a metric like a correlation to support your observation more quantitatively. The variations in discharge are very small, probably much smaller than the precision you get with the hydraulic estimate of discharge, so I am not convinced those results are robust enough. In addition, I guess your were walking into the river to do the velocity measurements, and this may have also been recorded in the seismic signal. Figure 11: Why is it needed to present the results for every 3 components of the seismic sensor?**

Response: Thank you for your suggestions. The three components of this instrument that monitors river processes are all related to river flow, which is more convincing.

**5. Results and Discussion**

**l. 397-403: Again, not clear from which experiment, test, location, section, etc. we are talking about.**

Response: Thank you for your suggestions. The specific description of the experiment has been modified in detail in the Methods section.

**l. 403-404: Not clear to me.**

Response: Thank you for your suggestions, this sentence has been modified to read as follows:' The total energy of the seismic waveforms generated by multiple sources is the sum of the energies in the river processes.'

**l. 466-472: In general, it looks like you are proposing an empirical model (and not a physical one as argued in the title) that requires calibration to predict discharge, while a physical model of turbulence has already been developed (Gimbert et al., 2014) and tested in multiple instances (e.g. Dietze et al., 2019), so I am not convinced by the usefulness of the proposed model.**

Response: Thanks. This study is a preliminary exploration of the use of seismic methods to monitor river processes, and an empirical model is proposed, which is an application of the physical model proposed by previous scholars.

**6. Conclusion**

**l. 487-503: The organization of the argument in this paragraph is very fuzzy.**

Response: Thank you for your suggestion. The last paragraph of the conclusion is rephrased and is shown in the manuscript.

**7. References**

**l. 516: Burtin et al. (2008) not cited in the manuscript, while it definitely should. l. 536: Foulds et al. (2014) not cited in the manuscript. Technical corrections**

Response: Thank you for your suggestions, the issue of these two references has been revised in the

literature.

**l. 176: "precision" instead of 'accuracy' I think.**
Response: Thank you for your suggestions, 'accuracy' has been changed to 'precision' in the manuscript.

**l. 510-586: in multiple instances (e.g. Schmandt et al., 2013), I saw that there are missing spaces between words in the references. Please check and correct throughout.**
Response: Thank you for your suggestions, the absence of spaces between words in the references has been revised.

**l. 563: Reference not sorted in alphabetic order.**
Response: Done, the references in the manuscript have been revised to be in alphabetical order.

**l.275: Tsai et al. (2012), and not Tasi. l. 568: Tsai et al. (2012) cited twice in the bibliography.**
Response: Done, it has been revised in the manuscript.

**References**

Anthony R E, Aster R C, Ryan S.: Measuring mountain river discharge using seismographs emplaced within the hyporheic zone, Journal of Geophysical Research: Earth Surface, 123(2): 210-228,2018.

Gimbert, F., Tsai, V.C., & Lamb, M. P. (2014). A physical model for seismic noise generation by turbulent flow in rivers. Journal of Geophysical Research: Earth Surface, 119, 1 – 30, doi:10.1002/2014JF003201.

Roth, D. L., Brodsky, E. E., Finnegan, N. J., Ricken man n, D., Turowski, J. M., & Ba doux, A. (2016). Bed load sediment transport inferred from seismic signals near a river. Journal of Geophysical Research: Earth Surface, 121(4), 725-747.

Schmandt, B., Aster, R. C., Scherler, D., Tsai, V. C., & Karlstrom, K. (2013). Multiple fluvial processes detected by riverside seismic and infrasound monitoring of a controlled flood in the Grand Canyon. Geophysical Research Letters,40, 4858–4863, doi:10.1002/grl.50953.

Tsai, V.C., Minchew, B., Lamb, M. P., & Ampuero, J.P. (2012). A physical model for seismic noise generation from sediment transport in rivers. Geophysical Research Letters, 39, n/a-n/a, doi:10.1029/2011GL050255.

Burtin, A., Bollinger, L., Vergne, J., Cattin, R., Nábělek, J.L., 2008. Spectral analysis of seismic noise induced by rivers: A new tool to monitor spatiotemporal changes in stream hydrodynamics. Journal of Geophysical Research: Solid Earth 113.https://doi.org/10.1029/2007JB005034.

Barrière, J., Oth, A., Hostache, R., & Krein, A.: Bedload transport monitoring using seismic observations in a low‐gradient rural gravel bed stream, Geophysical Research Letters, 42(7),2294–2301. https://doi.org/10.1002/2015GL063630,2015.

Hsu, L., Finnegan, N.J., Brodsky, E.E., 2011. A seismic signature of river bedload transport during storm events. Geophys. Res. Lett. 38 (13). https://doi.org/10.1029/2011GL047759

Herschy R.: The velocity-area method, Flow measurement and instrumentation, 4(1): 7-10,1993.

Díaz, J., Ruíz, M., Crescentini, L., Amoruso, A., &Gallart, J.: Seismic monitoring of an Alpine

mountain river, Journal of Geophysical Research: Solid Earth, 119(4),3276 – 3289. https://doi.org/10.1002/2014JB010955,2014.

Genç O, Ardıçlıoğlu M, Ağıralioğlu N.: Calculation of mean velocity and discharge using water surface velocity in small streams, Flow Measurement and Instrumentation,41: 115-120,2015.